# Impromptu VLA: Open Weights and Open Data for Driving Vision-Language-Action Models

Haohan Chi[*,1], Huan-ang Gao[*,1], Ziming Liu[†,2], Jianing Liu[1],
Chenyu Liu[1], Jinwei Li[1], Kaisen Yang[1], Yangcheng Yu[1], Zeda Wang[1], Wenyi Li[1],
Leichen Wang[2], Xingtao Hu[2], Hao Sun[2], Hang Zhao[3], Hao Zhao[1,†]

[1]AIR, Tsinghua University    [2]Bosch Research    [3]IIIS, Tsinghua University
[*]Equal contribution    [†]Corresponding author
Project Page: http://Impromptu-VLA.c7w.tech/

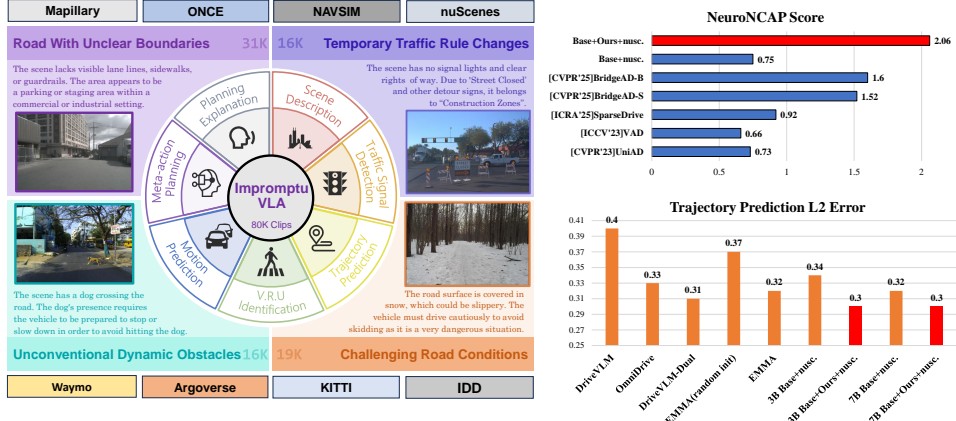

Figure 1: **Visual Abstract of Impromptu VLA.** We construct Impromptu VLA Dataset, which contains over 80K clips curated from 8 open-sourced datasets, focusing on four critical types of unstructured *"corner case"* scenarios that challenge current autonomous driving vehicles. It supports interconnected VLA tasks including scene understanding, prediction, meta planning and trajectory planning. Key experimental results demonstrates that VLA models trained with Impromptu VLA Dataset achieve significant performance improvements in both closed-loop and open-loop metrics.

## Abstract

Vision-Language-Action (VLA) models for autonomous driving show promise but falter in unstructured *corner case* scenarios, largely due to a scarcity of targeted benchmarks. To address this, we introduce Impromptu VLA. Our core contribution is the **Impromptu VLA Dataset**: over 80,000 meticulously curated video clips, distilled from over 2M source clips sourced from 8 open-source large-scale datasets. This dataset is built upon our novel taxonomy of four challenging unstructured categories and features rich, planning-oriented question-answering annotations and action trajectories. Crucially, experiments demonstrate that VLAs trained with our dataset achieve substantial performance gains on established benchmarks—improving closed-loop NeuroNCAP scores and collision rates, and reaching near state-of-the-art L2 accuracy in open-loop nuScenes trajectory prediction. Furthermore, our Q&A suite serves as an effective diagnostic, revealing clear VLM

improvements in perception, prediction, and planning. Our code, data and models are available at `https://github.com/ahydchh/Impromptu-VLA`.

# 1 Introduction

Autonomous driving has achieved remarkable progress, demonstrating increasing proficiency in navigating the well-structured environments of urban centers and highways where clear lane markings and predictable traffic flows are the norm [23, 29, 58]. However, the ultimate ambition of ubiquitous self-driving compels us to look beyond these well-trodden paths towards the intricate and often unpredictable domain of unstructured roads. These *unstructured* scenarios—encompassing everything from rural tracks and dynamic construction zones to areas with ambiguous signage or those recovering from natural events—represent the next significant frontier. It is here that current autonomous systems often face their sternest tests, and where breakthroughs are essential for realizing the full potential of go-anywhere autonomous capabilities [74].

Successfully navigating this frontier is profoundly hindered by a critical scarcity of specialized data. While numerous driving datasets have been foundational to current progress, they predominantly capture common, structured traffic situations [7, 8, 21, 42, 43, 55, 59, 68]. This leaves a significant blind spot concerning the sheer diversity and unique challenges posed by unstructured settings, such as ill-defined road boundaries, the appearance of unconventional dynamic obstacles, adherence to makeshift traffic rules, or dealing with treacherous road surfaces. Without large-scale, meticulously annotated datasets that specifically reflect these complex conditions [70, 47], the ability to train robust AI drivers and rigorously evaluate their adaptability in such scenarios remains severely constrained.

To address this data void, we introduce the **Impromptu VLA Dataset**, a new large-scale benchmark specifically curated to propel research in autonomous driving on unstructured roads, as introduced in Figure 1. Distilled from an initial pool of over two million clips from eight diverse public sources [7, 8, 21, 42, 43, 55, 59, 68], Impromptu VLA comprises approximately ∼80,000 meticulously selected and verified clips. These are categorized into four distinct types of challenging unstructured scenarios—roads with unclear boundaries, temporary traffic rule changes, unconventional dynamic obstacles, and challenging road conditions—and are enriched with extensive multi-task annotations and planning trajectories. The dataset was constructed using an advanced pipeline that leverages Vision-Language Models (VLMs) with Chain-of-Thought reasoning [39, 2, 12] for nuanced understanding, followed by comprehensive human verification to ensure high-quality, reliable labels.

Our comprehensive experimental evaluations rigorously validate the efficacy of the Impromptu VLA Dataset. We demonstrate that VLMs fine-tuned on our dataset exhibit substantially improved performance on established autonomous driving benchmarks. For instance, in challenging closed-loop NeuroNCAP [41] simulations (Table 2), our 3B model enhanced with Impromptu VLA saw its average NeuroNCAP score increase significantly to **2.06**/5.00 from 0.75/5.00 achieved by the baseline, while its average collision rate was critically reduced from 90.0% down to **65.1%**. In open-loop nuScenes [7] evaluations for trajectory prediction, pre-training with our dataset also markedly reduced L2 errors; our 3B model fine-tuned with Impromptu VLA achieved an average L2 error of 0.30m, bringing its performance nearly on par with leading specialized methods like EMMA+ [25] (0.29m), despite the latter often benefiting from substantially larger proprietary training datasets [88, 26]. Furthermore, evaluations on our dataset's own diverse Q&A validation suite reveal significant and quantifiable gains in specific VLM capabilities related to perception, prediction, and planning within these demanding unstructured contexts.

Our primary contributions are summarized as follows:

- The Impromptu VLA Dataset: A publicly available, large-scale, and richly annotated resource meticulously focused on diverse and challenging unstructured driving scenarios, designed to fill a critical gap in existing data resources.

- A systematic taxonomy for unstructured road conditions and a scalable, VLM-centric data curation pipeline for their identification, categorization, and comprehensive annotation with multi-task Q&A suitable for training advanced VLMs.

- Extensive experimental evidence demonstrating that training with the Impromptu VLA Dataset significantly boosts results on standard driving benchmarks, and serves as an

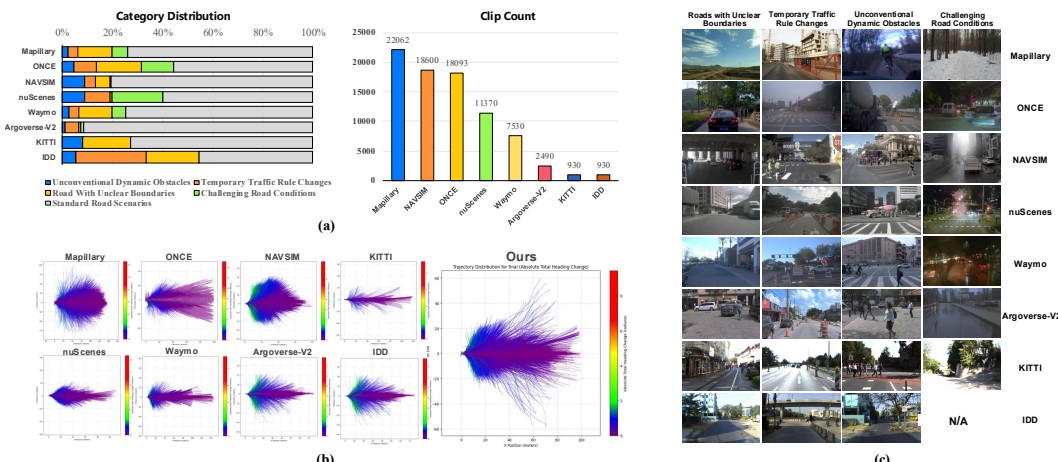

Figure 2: Characteristics comparison of different driving scene datasets. Figure (a) illustrates the distribution of scene categories across various datasets and the number of video clips contained in each category, providing a direct view of the emphasis on different scene types and the data scale of each dataset. Figure (b) compares the trajectory distribution in the original with the trajectory distribution in our constructed dataset, explaining the trajectory diversity of our dataset. Figure (c) shows examples of different scene categories from 8 source datasets. Notably, the IDD dataset lacks data for the "Challenging Road Conditions" category.

| Attribute | Mapillary | ONCE | NAVSIM | nuScenes | Waymo | Argoverse-V2 | KITTI | IDD | Sum. |
|---|---|---|---|---|---|---|---|---|---|
| Camera Views | 1 | 7 | 8 | 6 | 5 | 7 | 1 | 1 | - |
| Resolutions | Variable | 1920 x 1020 | 1920 x 1080 | 1600 x 900 | 640 x 360 | 1550 x 2048 | 1242 x 375 | 1920 x 1080 | - |
| FPS | 2Hz* | 2Hz | 2Hz | 2Hz | 10Hz | 20Hz | 10Hz | 15Hz | |
| Raw Clip Count | 1000k | 800k | 90k | 40k | 40k | 320k | 20k | 7k | 2000k |
| Labeled Clip Count | 22062 | 18093 | 18600 | 11370 | 7530 | 2490 | 930 | 930 | 80k |
| Raw Data Size | 60GB | 2TB | 450GB | 1.5TB | 5TB | 1TB | 180GB | 18GB | 10TB |
| Labeled Data Size | 1GB | 5GB | 12GB | 10GB | 7GB | 7GB | 1GB | 0.5GB | 43.5GB |

Table 1: **Dataset Information.** Summary of key attributes for the datasets used in this study. Note that the Mapillary dataset exhibits variable resolutions. For the Mapillary dataset, the frequency is assumed to be 2Hz as specific FPS information is not explicitly provided.

effective diagnostic tool for assessing and improving VLM capabilities in unstructured environments.

## 2 Impromptu VLA Dataset: Learning to Drive on Unstructured Roads

### 2.1 Overview

The research community currently lacks sufficient large-scale, diverse, and meticulously annotated datasets specifically focused on unstructured scenarios. To address this critical gap, we introduce the **Impromptu VLA Dataset**, a dataset curated to foster advancements in autonomous driving on unstructured roads. Sourced from an initial aggregation of over 2 million clips (occupying over 10T of storage) from eight prominent public datasets [7, 8, 21, 42, 43, 55, 59, 68], the Impromptu VLA Dataset has been distilled into a highly concentrated collection of ∼80,000 clips after our selection mechanism, which is illustrated in Figure 3. The resulting dataset specifically captures a diverse array of challenging scenarios, including roads with unclear boundaries, the presence of unconventional dynamic obstacles, and segments with temporary or non-standard traffic rules (see Table 1 for detailed statistics).

### 2.2 Defining a Taxonomy for Unstructured Driving Scenarios

A primary objective in creating the Impromptu VLA Dataset was to move beyond a monolithic and vague view of *unstructuredness* and establish a more granular understanding of the specific challenges these environments present. To achieve this, and to focus the dataset on scenarios that genuinely test

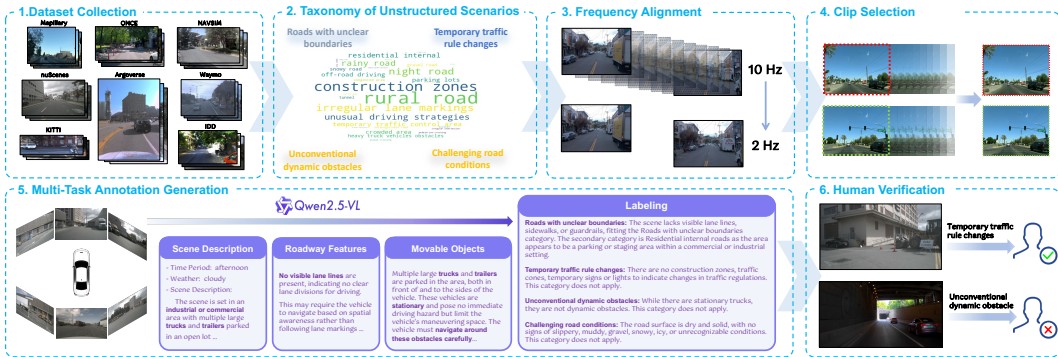

Figure 3: **Data Processing and Annotation Pipeline for the Impromptu VLA Dataset.** The diagram outlines the sequential process for creating our dataset, starting from raw data collection and scenario taxonomy definition (Sec. 2.2, through frequency alignment and keyclip selection, to multi-task annotation generation via Qwen2.5-VL (including scene description, object/feature analysis, and labeling), and concluding with rigorous human verification (Sec. 2.3).

the limits of current autonomous driving systems, our preliminary effort undertook a data-driven process to define a concise yet comprehensive taxonomy of unstructured road scenarios.

Our methodology for defining these categories began with an extensive, unbiased exploration of the collected data. We first created a representative subset by sampling approximately 10% of the clips at regular intervals from the aggregated and standardized multi-source dataset. This subset was then subjected to an open-ended descriptive analysis using the capabilities of a powerful Vision-Language Model, Qwen2.5-VL 72B [3]. Instead of querying the model to answer questions in a predefined label protocol, we leveraged the VLM's advanced image understanding capabilities to prompt it to generate detailed textual descriptions for each scene, as shown in the Appendix.

The subsequent phase involved a multi-stage, highly automated process to distill these descriptions into meaningful categories of unstructured challenges. First, to **programmatically identify and filter out conventional driving scenarios**, we employed another VLM-based classification step. Each initial, rich scene description generated by Qwen2.5-VL was evaluated using a carefully designed prompt, which instructed the VLM to act as a scenario categorizer to judge if the caption belongs to unconventional cases. To ensure the reliability and effectiveness of this VLM-based filtering prompt, we conducted an iterative refinement process of prompt. This process is tested on a validation subset of ~1000 scene descriptions, which were also manually and independently labeled as *'Conventional'* or *'Unconventional'* by two human annotators. The VLM's classifications were compared against human consensus, and the prompt was iteratively adjusted until achieving a high degree of agreement.

For the selected unconventional scenarios from the full set, we conduct semantic-level analysis to identify recurring patterns and group semantically similar unstructured scenarios. This clustering allowed for the bottom-up emergence of potential subcategories, such as those involving *"unclear road edges," "temporary road work," "animals on road,"* or *"poor visibility due to snow."* Through iterative refinement, consolidation of these machine-generated clusters, and abstraction based on the primary source of driving complexity identified in these groups, we converged on the four salient high-level categories detailed below.

*1. Roads with unclear boundaries*: Scenarios where the traversable path is ambiguous or undefined, such as rural dirt tracks, off-road trails, or roads with faded/absent markings. These severely challenge perception tasks like lane detection and drivable area segmentation.

*2. Temporary traffic rule changes*: Dynamic situations where standard traffic rules are temporarily altered by construction zones, human traffic controllers, or temporary signage, requiring autonomous vehicles to adapt to unusual instructions and road layouts.

*3. Unconventional dynamic obstacles*: Features dynamic actors or obstacles uncommon in typical urban driving that demand specialized interaction strategies. Examples include large or erratically moving vehicles, vulnerable road users in unexpected locations, or animal encounters, all posing sudden hazards.

*4. Challenging road conditions*: Encompasses scenarios where adverse road surfaces (e.g., potholes, mud, snow, ice) or environmental conditions (e.g., fog, heavy rain, low-light, glare) severely impair visibility or affect vehicle dynamics, complicating hazard perception and safe navigation.

## 2.3 Data Processing and Annotation

Following the definition of our unstructured scenario taxonomy (Section 2.2), the curated data underwent several processing and annotation stages as depicted in Figure 3.

**Keyclip Selection and Stability Filtering.** All collected sequences were first standardized to a uniform temporal rate of 2 Hz, addressing inconsistencies from diverse sources (Table 1). We aligned the clip configuration with NAVSIM [14], keeping 1.5 seconds from the past and 5 seconds for the future, and selected that central keyclip from each pack for annotation. To minimize false positives from transient keyclip-level predictions, we employed a temporal stability packing mechanism. Specifically, adjacent clips were packed into (up to if possible) 15-second "local-filter pack". Scene characteristic of a clip (preliminarily identified at keyclip-level) was only considered stable and propagated to subsequent annotation stages if it persisted for a minimum number of clips within this pack (e.g., more than a single occurrence). It is important to note that these "local-filter pack" were solely for this stability check and selection process; the final dataset primarily consists of individually annotated keyclips.

**Scene Classification and Structured Information Extraction via CoT prompting.** Selected keyclips were classified using Qwen2.5-VL 72B [3] with Chain-of-Thought (CoT) prompting [65] to extract rich structured information beyond simple captions. This hierarchical reasoning process analyzed overall scene context (R1: description), static roadway features (R2), movable objects (R3), and culminated in a justified final assignment (R4) to one of our four unstructured scene categories (Section 3.2). The structured CoT output provided not only the scene category but also a wealth of contextual information for subsequent task annotation.

**Multi-Task Annotation Generation.** Leveraging the scene category and the structured information extracted during the CoT process, we further enriched each keyclip with a diverse set of task-specific annotations, drawing inspiration from comprehensive annotation frameworks like Senna [28]. This multi-task annotation was achieved through a combination of rule-based and LLM-based methods. Specifically, we generated the following annotations for each selected keyclip. *1. Scene Description*: Comprehensive descriptions capturing the overall environmental context, time, weather, and traffic conditions were produced through targeted queries to VLM. *2. Traffic Signal Detection*: The presence state and type of active traffic signals were identified via further VLM queries. *3. Vulnerable Road User (VRU) Identification*: Information on VRUs, including their presence, type (e.g., pedestrian, cyclist), and distances from the ego vehicle, was derived from ground truth data. *4. Motion Intention Prediction*: To capture dynamic aspects, predicted motion intentions for key actors in the scene were generated by VLM. *5. Meta-action Planning*: High-level plans (e.g., accelerate-left, keep-straight) for the ego-vehicle were formulated, typically through VLM prompting conditioned on the scene context. *6. Planning Explanation*: Textual explanations, rationalizing potential or actual ego-vehicle maneuvers in response to the scene, were generated by the VLM. *7. End-to-End Trajectory Prediction*: Data to support this task was curated by structuring past vehicle states and corresponding future target trajectories in the ground truth.

**Comprehensive Human Verification.** All generated annotations—both the primary unstructured scene category and the subsequent multi-task labels—were subjected to a meticulous human verification process. Annotators reviewed each keyclip and its associated labels, providing a binary judgment (accept/reject) or performing minor corrective edits if necessary. This ensured high fidelity across the entire dataset. To quantitatively assess the VLM's scene classification performance for our defined unstructured categories prior to extensive human review, we evaluated it on a subset of 200 images sampled at intervals from the nuScenes dataset. Comparing VLM classifications against expert manual labels yielded strong F1 scores for several categories: 0.90 for 'Temporary Traffic Rule Changes', 0.81 for 'Unconventional Dynamic Obstacles', and 0.91 for 'Challenging Road Conditions'. The 'Road With Unclear Boundaries' category was found to be too rare within this specific nuScenes subset for a meaningful F1 score calculation. These validation results provide confidence in the VLM-based stages of our annotation pipeline.

| Source | Method | NeuroNCAP Score ↑ | | | | Collision rate (%) ↓ | | | |
|---|---|---|---|---|---|---|---|---|---|
| | | Avg. | Stat. | Frontal | Side | Avg. | Stat. | Frontal | Side |
| CVPR 2023 | UniAD[2] | 0.73 | 0.84 | 0.10 | 1.26 | 88.6 | 87.8 | 98.4 | 79.6 |
| ICCV 2023 | VAD[2] | 0.66 | 0.47 | 0.04 | 1.45 | 92.5 | 96.2 | 99.6 | 81.6 |
| ICRA 2025 | SparseDrive[1] | 0.92 | - | - | - | 93.9 | - | - | - |
| CVPR 2025 | BridgeAD-S[1] | 1.52 | - | - | - | 76.2 | - | - | - |
| CVPR 2025 | BridgeAD-B[1] | 1.60 | - | - | - | 72.6 | - | - | - |
| - | Base+nuScenes | 0.75 | 0.99 | 0.55 | 0.70 | 90.0 | 88.6 | 93.2 | 88.0 |
| - | **Base+Impromptu+nuScenes** | **2.06** | **2.55** | **1.86** | **1.78** | **65.1** | **54.8** | **72.8** | **67.6** |

Table 2: Results on NeuroNCAP. (where [1] indicates sourced from [81] and [2] indicates sourced from [40]) Best scores in each category (without/with post-processing) are in **bold**, second best are underlined. The improvements in both the overall NeuroNCAP score and, crucially, the reduction in collision rates suggest that our dataset helps the model develop a more nuanced understanding of complex road interactions, leading to more robust and safer driving policies.

To provide a clearer measure of this process and the initial VLM performance, we logged detailed statistics during verification. The initial acceptance rate for VLM-generated annotations by our human annotators was approximately 81%. For the remaining 19% of annotations that required review, 65% were ultimately discarded as they did not meet our criteria for unstructured scenarios, while 35% were corrected by human annotators. A breakdown of the rejected 19% across the four categories reveals where the VLM faced the most challenges: 'Challenging Road Conditions' accounted for 42.5% of rejections, followed by 'Unconventional Dynamic Obstacles' (32.0%), 'Temporary Traffic Rule Changes' (22.9%), and 'Road With Unclear Boundaries' (2.6%). These figures quantify the VLM's initial accuracy and underscore the critical role of our human verification phase in ensuring the final dataset's high fidelity.

## 2.4 Dataset Statistics

The final Impromptu VLA Dataset comprises a substantial collection of annotated clips specifically curated for their unstructured road characteristics. Figure 2 illustrates the total number of these clips derived from each source dataset and presents the overall distribution of these clips across the four defined unstructured scenario categories introduced in Sec. 2.2. The coverage of trajectory distribution is also reported in Figure 2.

To maximize the utility of this dataset for training and evaluating perception and planning models, the rich multi-task annotations generated for each clip (as detailed in Sec. 2.3) are structured as planning-oriented Question-Answering (Q&A) pairs. This format, inspired by frameworks like DriveVLM [58] or EMMA [25], directly associates visual inputs, text outputs and action trajectory predictions within sequence space of LLMs. For standardized evaluation, the entire dataset of curated clips, across all four unstructured categories, is partitioned into training and validation sets using an 80:20 split. This stratification is performed within each category to ensure that the validation set maintains a representative distribution of all defined unstructured road challenges.

## 3 Experiments

This section empirically validates our Impromptu VLA Dataset by investigating its impact on advancing autonomous driving models. We seek to answer:

(1) Does training with our dataset improve vision-language model (VLM) performance on existing benchmarks, both closed-loop and open-loop?

(2) In which specific aspects does the Impromptu VLA Dataset enhance VLM performance - perception, prediction, or planning? How effectively does our validation set, with its detailed planning-oriented Q&A, serve as a diagnostic benchmark for pinpointing these contributions and evaluating model capabilities in these distinct tasks?

### 3.1 Pushing Forward Boundaries of Existing End-to-end Autonomous Driving Benchmarks

**Closed-loop evaluation.** We choose NeuroNCAP [41], a comprehensive closed-loop evaluation framework that leverages the nuScenes dataset to simulate a wide array of challenging real-world driving scenarios, allowing for the assessment of an autonomous vehicle's planning and control systems in terms of safety and efficiency under diverse conditions. The NeuroNCAP evaluation quantifies performance primarily through collision rates and the NeuroNCAP score (NNS). The NNS is computed, in the spirit of a 5-star rating system, as follows: a score of $5.0$ is achieved if no collision occurs; otherwise, the score is $4.0 \cdot \max(0, 1 - v_i/v_r)$, where $v_i$ is the actual impact speed (magnitude of relative velocity between the ego-vehicle and the colliding actor) and $v_r$ is the reference impact speed that would occur if no evasive action were performed. This means that if a collision is not avoided, the score linearly decreases from a potential 4 points towards 0 as the impact speed $v_i$ approaches or exceeds the reference speed $v_r$. Collision rates, on the other hand, directly track the percentage of scenarios resulting in a collision. These two metrics are categorized by interaction types (e.g., frontal, side).

Our method involves a comparative study of two distinct training pipelines. The base model here is Qwen2.5VL 3B [3]. The first pipeline, which we term **"Base+Impromptu+nuScenes"** in Table 2, involves initially fine-tuning the base VLM on the training split

| Method | L2 Error (m) ↓ | | | |
|---|---|---|---|---|
| | 1s | 2s | 3s | **Avg.** |
| *Closed-source API-only Models* | | | | |
| GPT-4o[1] [24] | **0.28** | **0.93** | **2.02** | **1.07** |
| Claude-3.5-Sonnet[1] | 0.29 | 0.98 | 2.12 | 1.13 |
| Claude-3.7-Sonnet[1] | **0.28** | 0.94 | 2.04 | 1.09 |
| Gemini-2.0-Flash[1] | 0.31 | 1.08 | 2.36 | 1.25 |
| Gemini-2.5-Pro[1] | 0.37 | 1.35 | 2.96 | 1.56 |
| *Open-source Generalist VLMs* | | | | |
| LLaVA-1.6-Mistral-7B[2] | 1.49 | 3.38 | 4.09 | 2.98 |
| Llama-3.2-11B-Vision-Instruct[2] | 1.54 | 3.31 | 3.91 | 2.92 |
| Qwen2-VL-7B-Instruct[2] [80] | 1.45 | 3.21 | 3.76 | 2.81 |
| DeepSeek-VL2-16B[1] [71] | 0.66 | 1.68 | 2.92 | 1.75 |
| DeepSeek-VL2-28B[1] [71] | **0.37** | 1.35 | 2.96 | 1.56 |
| LLaMA-3.2-11B-Vision-Instruct[1] | 0.52 | 1.42 | 2.68 | 1.54 |
| LLaMA-3.2-90B-Vision-Instruct[1] | 0.66 | 1.71 | 3.01 | 1.79 |
| Qwen-2.5-VL-7B-Instruct[1] [4] | 0.46 | **1.33** | **2.55** | **1.45** |
| *Training-based Driving Specialists (Existing Methods)* | | | | |
| UniAD[3] [23] | 0.42 | 0.64 | 0.91 | 0.66 |
| VAD[3] [29] | 0.17 | 0.34 | 0.60 | 0.37 |
| BEV-Planner[3] [37] | 0.16 | **0.32** | **0.57** | **0.35** |
| Ego-MLP[3]* [37] | **0.15** | **0.32** | 0.59 | **0.35** |
| *Ours and Key Competitors (Specialized Driving Models)* | | | | |
| DriveVLM[3] [58] | 0.18 | 0.34 | 0.68 | 0.40 |
| OmniDrive[3] [61] | 0.14 | 0.29 | 0.55 | 0.33 |
| DriveVLM-Dual[3] [58] | 0.15 | 0.29 | **0.48** | 0.31 |
| EMMA (random init) [25][3] | 0.15 | 0.33 | 0.63 | 0.37 |
| EMMA [25][3] | 0.14 | 0.29 | 0.54 | 0.32 |
| EMMA+[3] [25] | 0.13 | 0.27 | 0.48 | 0.29 |
| 3B Base+nuScenes | 0.14 | 0.30 | 0.58 | 0.34 |
| 3B Base+Impromptu+nuScenes | **0.13** | **0.27** | 0.52 | **0.30** |
| 7B Base+nuScenes | **0.13** | 0.28 | 0.55 | 0.32 |
| 7B Base+Impromptu+nuScenes | **0.13** | **0.27** | 0.51 | **0.30** |
| 11B Base+nuScenes | **0.13** | 0.28 | 0.54 | 0.32 |
| 11B Base+Impromptu+nuScenes | **0.13** | **0.27** | 0.51 | **0.30** |

Table 3: Open-loop trajectory prediction L2 errors (m) on the nuScenes dataset. (where [1] indicates sourced from [48], [2] indicates sourced from [75] and [3] indicates sourced from [25]). Best results within each category are in **bold**, second best are underlined.

of our Impromptu VLA dataset, and subsequently further fine-tuning this adapted model on the nuScenes training set. The second pipeline, **"Base+nuScenes"**, directly fine-tunes the base VLM on the nuScenes training set without any exposure to the Impromptu VLA. Both models are then evaluated on the NeuroNCAP benchmark.

**Open-loop Evaluation.** In addition to closed-loop simulations, we conduct open-loop evaluations to specifically assess the trajectory prediction accuracy of VLMs when benefiting from our Impromptu VLA. For this, we also utilize the nuScenes dataset [7], focusing on the end-to-end trajectory prediction task. Performance is primarily measured by the L2 distance (in meters) between the predicted and ground truth trajectories at future time horizons of 1s, 2s, and 3s, along with the average L2 error. The experimental methodology mirrors the comparative approach used in the closed-loop tests. We compare two main training strategies for three VLM bases: Qwen2.5VL 3B, Qwen2.5VL 7B, and LLaMA-3.2-11B-Vision-Instruct: (1) **"Base+nuScenes"**, where base VLM is directly fine-tuned on the nuScenes dataset, and (2) **"Base+Impromptu+nuScenes"**, where base VLM is first fine-tuned on our Impromptu VLA, and this adapted model is then further fine-tuned on nuScenes. This comparison aims to isolate the benefits conferred by pre-training on our dataset for the task of trajectory prediction in diverse scenarios. It should be noted that due to computational

constraints, the reported metrics are the result of a single run. The results, contextualized with several state-of-the-art methods, are detailed in Table 3.

As demonstrated in Table 3, the open-loop trajectory prediction results on the nuScenes benchmark reveal a marked improvement when models are pre-trained on our Impromptu VLA Dataset. The gains in trajectory accuracy are consistently observed across the 1s, 2s, and 3s prediction horizons. Impressively, this level of enhancement brings the performance of our adapted 3B/7B/11B models, into a competitive range with leading methods such as EMMA+ [25] (average L2 of 0.29m), despite EMMA+ benefiting from training on substantially larger internal datasets with millions of scenarios, introduced by Waymo. Crucially, the results from the LLaMA-3.2-11B model confirm that these benefits are generalizable across different VLM architectures. This underscores the efficacy of the Impromptu VLA Dataset (80K clips) in significantly boosting trajectory prediction capabilities.

## 3.2 Diagnostic Evaluation of VLM Capabilities on Impromptu VLA

To answer the second question—investigating which specific aspects of autonomous driving (perception, prediction, or planning) are enhanced by the Impromptu VLA Dataset and how well our validation set serves as a diagnostic benchmark—we conducted a series of evaluations using its planning-oriented Q&A tasks. This involved comparing the performance of base Vision-Language Models (VLMs) against versions fine-tuned on our dataset in a task-oriented manner.

| Method | Q&A Accuracy ↑ | | | | Traj. Pred. L2 Error (m) ↓ | | | | |
|---|---|---|---|---|---|---|---|---|---|
| | V.R.U. | T. Light | Dyn. Obj. | M.P. | 1s | 2s | 3s | 4s | Avg. |
| 3B Base | 0.87 | 0.95 | 0.20 | 0.56 | 3.39 | 5.31 | 7.70 | 10.08 | 6.62 |
| 3B Base+nuScenes | 0.90 | 0.96 | 0.75 | 0.72 | 1.12 | 1.62 | 2.24 | 3.00 | 2.00 |
| 3B Base+Impromptu | **0.91** | **0.96** | **0.92** | **0.84** | **0.16** | **0.43** | **0.82** | **1.34** | **0.69** |
| 7B Base | 0.86 | 0.92 | 0.22 | 0.55 | 2.99 | 4.80 | 6.64 | 8.52 | 5.74 |
| 7B Base+Impromptu | **0.91** | **0.97** | **0.92** | **0.83** | **0.10** | **0.31** | **0.65** | **1.11** | **0.54** |

Table 4: **Quantitative Evaluation on the Impromptu VLA validation set.** Performance comparison on various QA tasks within our validation set. The table shows metrics for 3B and 7B Qwen2.5-VL models. Accuracy ↑ is reported for perception (V.R.U., T. Light), prediction (Dyn. Obj.), meta-planning (M.P.) and Planning (L2). Best results are in **bold**.

The quantitative evaluation on the Impromptu VLA validation set, summarized in Table 4, reveals the distinct advantages of our dataset. To provide a fair comparison, we evaluated the 3B Base+nuScenes model as a strong baseline, which has been fine-tuned on a standard driving dataset. As shown, the 3B Base+nuScenes model clearly outperforms the 3B Base (e.g., 2.00m vs 6.62m avg. L2 error), confirming that training on nuScenes data provides substantial end-to-end trajectory prediction capabilities.

However, the 3B Base+Impromptu model, trained on our dataset, demonstrates significantly superior performance, surpassing the 3B Base+nuScenes model across most metrics. Most notably, the average L2 trajectory error drops from 2.00m to just 0.69m. This gap highlights the diversity and richness of our Impromptu VLA dataset. As depicted in Figure 2(b), nuScenes trajectory data tends to be more concentrated, while our dataset features a wider variety of trajectories. This diversity enables models trained on Impromptu to generalize better to more varied driving scenarios. Furthermore, the improvements in other tasks like 'Dyn. Obj.' and 'M.P.' suggest that training with our planning-oriented data enhances the model's overall scene understanding, implying a strong connection between trajectory learning and broader descriptive and reasoning capabilities.

## 4   Related Works

**When Vision Language Models Meet Autonomous Driving.** Vision Language Models (VLMs) extend Large Language Models (LLMs) with visual understanding capabilities [39, 2, 12, 63, 1, 17, 31, 32, 76, 87, 54, 22, 11, 84, 67, 82, 36, 72, 86, 30, 85, 79, 34] enabling multimodal reasoning. These models have recently been introduced into autonomous driving, either to complement traditional end-to-end frameworks [13, 23, 29, 11, 46, 83] or to function as standalone decision-makers [77, 10, 49, 25, 58], as they are assumed to be able to transfer the generalization ability to the road scenes [1, 62, 50, 52, 60, 44, 51]. Furthermore, novel approaches leverage collaborative LLM-agents for

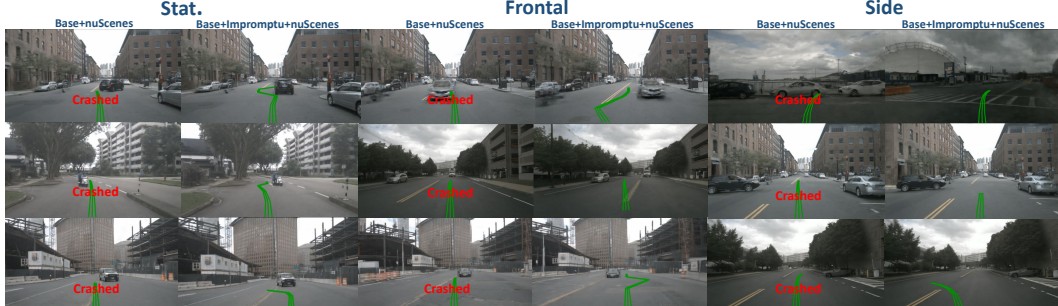

Figure 4: **NeuroNCAP performance in challenging scenarios.** This figure compares the driving behavior of the two models in three representative challenging scenarios: static, frontal, and side. For each scenario, the left column shows the behavior of the base model, which is fine-tuned on nuScenes. The right column shows the performance of the model trained on a subset of our proposed dataset and then fine-tuned on nuScenes. Compared to the base model, the model using our data can better avoid vehicles by turning, slowing down, etc.

editable scene simulation, offering new paradigms for data generation. [66] In this line of research, some approaches translate structured driving inputs—such as perception outputs and HD maps—into language for planning [10, 49], while others like DriveGPT4 [77] process front-camera video to predict both control commands and rationales. LVLM-based planners have also been validated in simulation environments such as CARLA [19, 64], and large-scale pretraining (e.g., ELM [88]) has shown promise for improving generalization. Recent works further propose driving-specific Q&A data and benchmarks [53, 70, 47, 69, 15, 33] to better align training with downstream planning tasks.

**Specialized Techniques and Datasets for Autonomous Driving.** Beyond the VLM paradigm, significant research continues to advance various critical aspects of autonomous driving systems, addressing specific challenges in perception, simulation, mapping, and prediction. For instance, in the realm of realistic simulation, Mars [72] offers an instance-aware, modular, and realistic simulator leveraging neural radiance fields, crucial for generating and testing complex scenarios. Challenger [78] focuses on generating physically plausible yet realistic adversarial driving videos to stress-test AD systems against aggressive maneuvers, while AVD2 [35] introduces a novel framework for generating accident videos aligned with detailed natural language descriptions and preventative measures, thereby enhancing accident scenario understanding for training and analysis. To enhance robustness in challenging environmental conditions, especially at night, joint self-supervised nighttime image enhancement and depth estimation are explored by Steps [86], crucial for improved visual perception in low-light settings. Accurate and far-reaching environmental representation is further enabled by P-MapNet [30], which leverages both standard definition (SDMap) and high-definition (HDMap) priors for superior map generation, improving situational awareness over longer distances. In the domain of 3D scene understanding from limited inputs, MonoOcc [85] delves into monocular semantic occupancy prediction, aiming to reconstruct comprehensive 3D geometry and semantics from single-camera views. For robust motion forecasting, especially in dynamic multi-agent environments, Int2 [79] presents a large-scale dataset and framework specifically for interactive trajectory prediction at complex intersections, capturing crucial dynamics that are vital for safe navigation. These targeted innovations collectively pave the way for more capable and reliable autonomous driving systems. Efforts also extend to generating high-quality, annotated training data, with UniScene [34] proposing a unified occupancy-centric framework for comprehensive driving scene generation. Furthermore, SCP-Diff [20] significantly improves the quality of semantic image synthesis for sensor simulation by introducing a spatial-categorical joint prior, enabling the creation of highly realistic and diverse virtual environments with precise semantic control. These targeted innovations collectively pave the way for more capable and reliable autonomous driving systems.

**End-to-end Autonomous Driving Datasets and Benchmarks.** We categorize autonomous driving benchmarks into two categories, one for large-scale imitation learning, and one for simulation. The first category includes large-scale, real-world datasets, often collected from road networks [7, 8, 42, 43, 55, 59, 68], which are crucial for developing and evaluating systems on annotated perception, prediction, and planning tasks. In this work, we select representative imitation learning benchmarks for constructing our dataset: KITTI [21], an early benchmark, provided data from

Germany. nuScenes [7] expanded on this with data from Boston and Singapore. The Waymo Open Dataset [55] offers immense scale with data collected from diverse US locations. Argoverse (v1 & v2) [9, 68] also features data from various US cities. nuPlan provides over 1200 hours of driving data from cities in the US and Singapore. For global visual diversity, Mapillary Vistas [43] includes street-level imagery from all continents. ONCE [42] contributes a massive dataset with 1 million LiDAR scenes and 7 million camera images from China. Finally, the India Driving Dataset provides crucial data from challenging and unstructured driving environments across India.[59, 18] The second line involves simulation-based benchmarks, such as Bench2Drive [27], NAVSIM [14], and NeuroNCAP [41], which offer closed-loop evaluation environments. These simulators utilize metrics more akin to driving task-oriented reward designs, allowing for systematic testing of decision-making and control algorithms in interactive scenarios. Notably, our dataset construction prioritizes the collection and filtering of authentic, real-world unstructured scenarios, rather than introducing synthetic elements or anomalies [57, 6, 56, 16, 38, 5, 45, 73],. This commitment to genuine data ensures the Impromptu VLA Dataset fosters the development of VLA models grounded in the true complexities of diverse driving conditions.

## 5    Conclusion

This paper introduced the Impromptu VLA Dataset, a meticulously curated benchmark of approximately 80,000 clips featuring rich multi-task Question-Answering annotations and corresponding action trajectories, specifically designed to address the critical data scarcity for autonomous driving in unstructured environments. Our comprehensive experiments demonstrate that Vision-Language Models trained with the Impromptu VLA Dataset achieve significant performance gains, evidenced by enhanced closed-loop safety and driving scores on the NeuroNCAP benchmark, as well as improved open-loop trajectory prediction accuracy on nuScenes. Furthermore, evaluations on our dataset's validation suite confirm its efficacy as a diagnostic tool, revealing specific model advancements in perception, prediction, and planning capabilities when handling diverse and challenging unstructured road scenarios. The Impromptu VLA Dataset thus offers a valuable new resource to foster the development of more robust, adaptable, and capable autonomous driving systems prepared for the complexities of real-world operation.

**Contribution in Context and Future Outlook.** We position the Impromptu VLA Dataset as a critical step in the iterative process of achieving truly intelligent driving. The field has evolved from mastering foundational capabilities in simple scenes , and our work contributes by systematically expanding coverage to the long-tail distribution of unstructured scenarios . We recognize that targeting unstructured environments requires specific and careful design, especially when curating from existing sources. Furthermore, we place trust in the emergent capabilities of advanced models and hypothesize that by training on our diverse, curated data, models will develop the potential to generalize to novel unstructured situations not explicitly seen during training. Ultimately, a primary contribution of this paper is to define and formalize the challenges posed by these unstructured scenarios, rather than to offer a complete, one-size-fits-all solution. We eagerly anticipate and welcome subsequent research that will build upon this foundation to further advance the field.

**Limitation.** We acknowledge that the primary reliance on Qwen2.5-VL for annotation generation in the Impromptu VLA Dataset might introduce potential model-specific biases; however, we believe the holistic human verification and the demonstrated utility in enhancing Vision-Language Model performance in unstructured scenarios confirms its significant value as a research resource.

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

# Appendix

This supplementary document provides additional insights and technical details regarding our proposed Impromptu VLA dataset and its associated models. We will begin by detailing the implementation specifics of our training process, including data formats, hyperparameter settings, and computational costs (Section A). Subsequently, we will elaborate on our construction for the dataset of unstructured driving scenarios (Section B). We then present two key ablation studies: first, a validation of our Chain-of-Thought (CoT) prompting strategy (Section C), and second, an analysis of the contributions of different QA task types within our dataset (Section D). Finally, we will offer concrete Question-Answering (Q&A) examples from the Impromptu VLA dataset to illustrate its structure (Section E).

## A   Implementation Details

In this section, we provide further details regarding the training processes, including the data format specifically employed when fine-tuning on the nuScenes dataset, and the general hyperparameter settings used for the Qwen2.5-VL model variants during our experiments.

### A.1   Data format for nuScenes Fine-tuning and Evaluation

The data format described here pertains specifically to the fine-tuning and evaluation stages performed directly on the nuScenes dataset, as referenced in our main paper's open-loop and closed-loop experiments. While the trajectory data format shares similarities with the End-to-End Trajectory Prediction Q&A format within our Impromptu VLA dataset (which uses past 1.5s and future 5s trajectories), there are distinctions for this nuScenes-specific setup. These include utilizing only the front-camera view and using trajectories spanning the past 3 seconds and future 3 seconds. For this nuScenes-specific training, input consists of the ego-vehicle's past trajectory points. For testing on nuScenes trajectory prediction benchmarks, in addition to past trajectory, velocity, and acceleration (similar to our Impromptu VLA Q&A format), we also incorporate steering wheel angle information where available from the nuScenes dataset. An example of the data format used for nuScenes experiments is illustrated in Figure 5.

### A.2   Hyperparameters

Our empirical hyperparameter values, outlined in Table 5, were established for training models using the Impromptu VLA Dataset and subsequently on nuScenes data. These values stem from general observation and common practices, not a thorough optimization process, due to the extensive search space. Nevertheless, with these settings, our models demonstrate a superior grasp of unstructured environments, leading to a significant improvement in trajectory prediction, as shown in the main paper. We recognize that further refinement through exhaustive tuning could yield even better results.

| Hyperparameter | Value |
|---|---|
| Cutoff len | 4096 |
| Finetuning type | full |
| Image resolution | 262144 |
| Learning rate | 5.0e-06 |
| Scheduler type | cosine |
| Warmup ratio | 0.03 |

Table 5: **Hyperparameters used in training**. This table details the empirical values for crucial hyperparameters employed across our entire pipeline. These settings were consistently applied in all experiments and were derived from general observations rather than specific task-based tuning.

## B   VLM-based Unstructured Scene Identification

Following the Data Exploration phase where rich textual descriptions were generated for each scene, our objective was to isolate the scenarios that genuinely represented "unstructured" or "unconven-

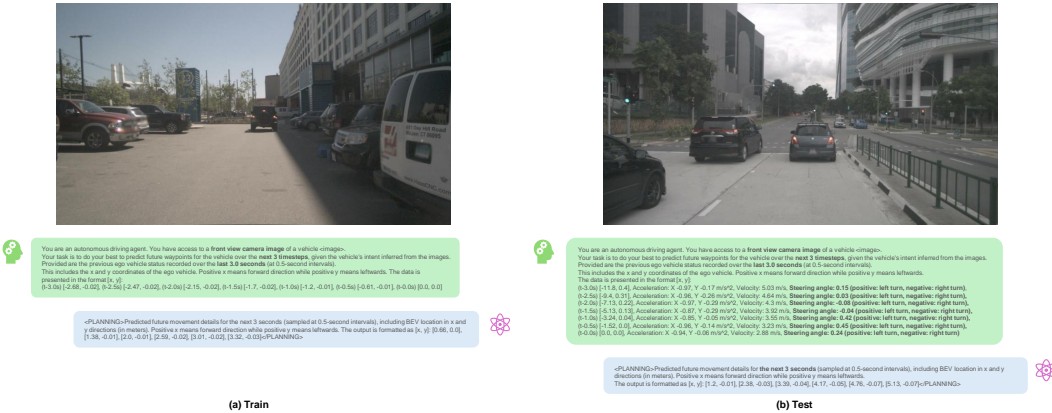

(a) Train        (b) Test

Figure 5: Example of data used when training and testing on nuScenes

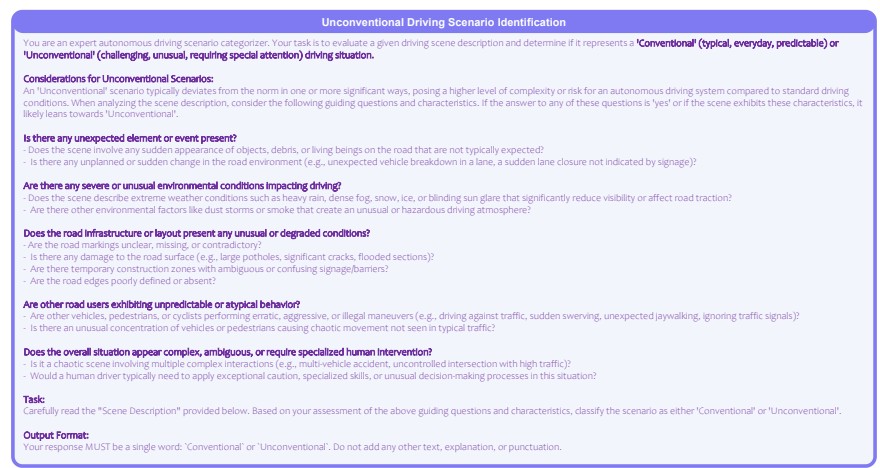

Figure 6: Prompt used during Unconventional Driving Scenario Identification

tional" driving conditions. This was critical for focusing the dataset on true corner cases. To achieve this, we employed a VLM-based classification step. Each scene description generated by Qwen2.5-VL was evaluated using a carefully designed prompt, which instructed the VLM to act as a scenario categorizer and determine if the description corresponded to an unconventional case (as opposed to a routine, structured driving situation).

The effectiveness of this VLM-based filtering prompt was crucial. Therefore, we conducted an iterative refinement process on a validation subset of approximately 1000 scene descriptions, which were also independently labeled by two human annotators as 'Conventional' or 'Unconventional'. The VLM's classifications were compared against this human consensus, and the prompt (an example of which is shown in Figure 6) was iteratively adjusted until a high degree of agreement was achieved. This ensured that the scenarios passed to the next stage were indeed representative of the complex conditions we aimed to capture.

## C    Ablation Study on Prompting Strategies

To validate the effectiveness of our proposed Chain-of-Thought (CoT) prompting strategy, we conducted a comparison of our CoT (Chain-of-Thought) prompting strategy against a simpler flat

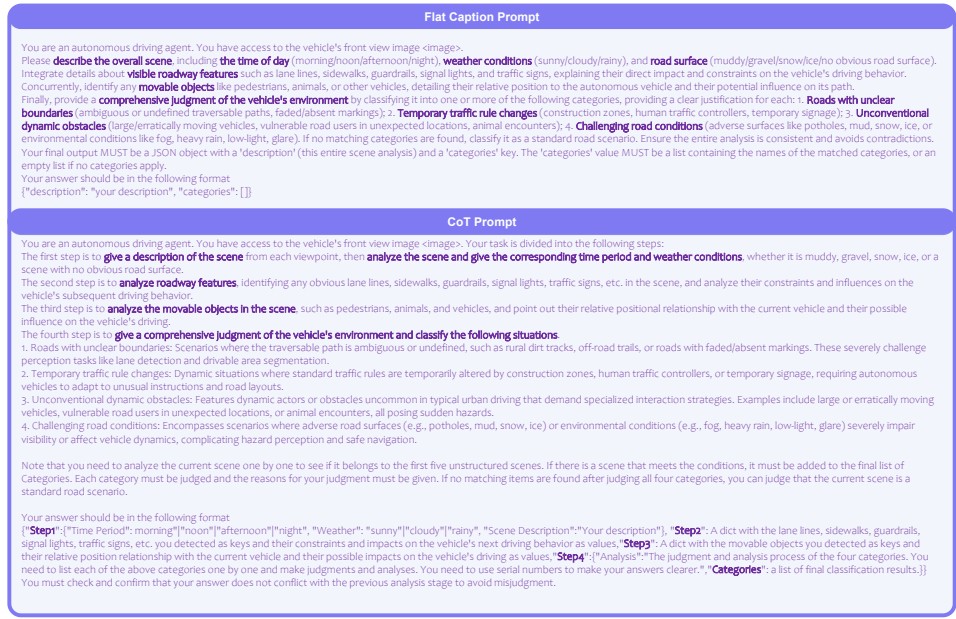

Figure 7: Different prompting strategies

captioning approach on a subset of the Mapillary dataset, comprising 1000 images. The two prompts used for this comparison are shown in Figure 7. The results demonstrate that our CoT strategy indeed improves classification accuracy.

Specifically, the precision for flat captioning was 0.63, while our CoT approach achieved a precision of 0.70. This indicates that the more structured reasoning facilitated by CoT leads to more accurate categorization.

Table 6: Ablation study on the Impromptu VLA validation set (using the 3B model). Each row shows the performance when the specified QA data type was **excluded** from the training set.

| QA data types not used for training | Q&A Accuracy ↑ | | | | Traj. Pred. L2 Error (m) ↓ | | | | |
|---|---|---|---|---|---|---|---|---|---|
| | V.R.U. | T. Light | Dyn. Obj. | M.P. | 1s | 2s | 3s | 4s | Avg. |
| V.R.U | 0.91 | 0.96 | 0.92 | 0.83 | 0.13 | 0.37 | 0.74 | 1.23 | 0.62 |
| Dyn. Obj. | 0.92 | 0.96 | **0.66** | 0.83 | 0.13 | 0.38 | 0.76 | 1.25 | 0.63 |
| Plan Exp. | 0.92 | 0.96 | 0.92 | 0.84 | 0.13 | 0.38 | 0.75 | 1.24 | 0.63 |
| T. Light | 0.92 | 0.96 | 0.92 | 0.84 | 0.11 | 0.35 | 0.71 | 1.19 | 0.59 |
| Scene Des. | 0.92 | 0.96 | 0.92 | 0.84 | 0.11 | 0.36 | 0.73 | 1.22 | 0.61 |
| M.P. | 0.92 | 0.96 | 0.92 | **0.66** | 0.11 | 0.34 | 0.70 | 1.17 | 0.58 |
| Traj. Pred. | 0.92 | 0.96 | 0.92 | 0.84 | 3.37 | 5.72 | 7.99 | 10.54 | 6.91 |

# D   Ablation Study on QA Task Contributions.

To further understand the contribution of different data types within our dataset and verify the impact of data richness, we conducted a series of ablation experiments. We used the 3B Base+Impromptu model as our baseline and systematically removed one category of QA data at a time during training, evaluating the model's performance on the full validation set. The results are presented in Table 6.

We observed that when Dyn. Obj. or M.P. QA data were excluded from training, the model's performance on those specific tasks (0.66 and 0.66 respectively) was still significantly higher than the 3B Base model's performance (0.20 and 0.56, respectively, from Table 4). This suggests that other QA data categories contribute positively to the model's performance on these specific tasks, even when their direct supervision is removed. For example, improved scene understanding derived from training on trajectory or description data might indirectly benefit other perception-based tasks. As expected, the most critical component for planning is the trajectory data itself; removing Traj. Pred. data caused the average L2 error to skyrocket from 0.69m to 12.97m. This highlights that our

dataset serves as a robust diagnostic tool, effectively revealing the model's learned capabilities and the complex interplay between different types of driving knowledge.

# E   Q&A Examples

Our Question-Answering (QA) format for the Impromptu VLA dataset, inspired by frameworks like Senna, is designed to be comprehensive. We present two distinct QA data examples from various source datasets that contribute to Impromptu VLA, as illustrated in Figures 8 through 16. These examples include the corresponding front-facing images and detailed QA information.

A notable aspect of our QA format is the inclusion of special tokens to differentiate between tasks with similar output structures. For instance, tokens like `<PLANNING>` and `<DYNAMIC_OBJECTS>` are used. We found through experimentation that without such disambiguation, the model could confuse, for example, the ego-vehicle's meta-action plan with the predicted motion intentions for other dynamic objects in the scene, due to the structural similarity of their speed and path plan outputs. These special tokens ensure the model can distinguish the context effectively.

The Impromptu VLA dataset includes an end-to-end trajectory prediction Q&A task. As detailed in the main paper, this incorporates the ego-vehicle's past 1.5 seconds of states (including displacement, velocity, and acceleration relative to its current position) and its ground-truth future 5-second trajectory. This formulation is designed to facilitate the model's learning of long-range trajectory prediction, a key capability that also supports tasks like understanding and executing navigation-related commands.

Additionally, it is important to note that the availability of all defined Q&A tasks for a given clip in the Impromptu VLA dataset depends on the richness of the annotations in the source dataset from which the clip was curated. For instance, if a source clip lacks ground-truth object detection data, generating Q&A pairs for tasks like Vulnerable Road User (V.R.U.) Identification (which relies on ground truth [cite: 82]) or detailed Motion Intention Prediction for specific actors may not be possible for that particular clip, even if other Q&A tasks like scene description or ego-trajectory prediction can be generated.

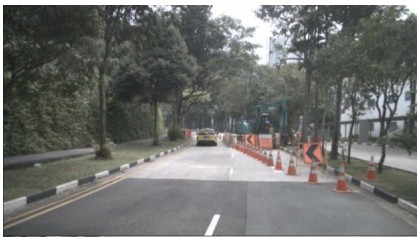

**Scene Description:**
**Q:** Suppose you are driving, and I'm providing you with the image captured by the car's front camera, generate a description of the driving scene which includes the key factors for driving planning, including the positions and movements of vehicles and pedestrians; prevailing weather conditions; time of day, distinguishing between daylight and nighttime; road conditions, indicating smooth surfaces or the presence of obstacles; and the status of traffic lights which influence your decision making, specifying whether they are red or green. The description should be concise, providing an accurate understanding of the driving environment to facilitate informed decision-making.

**A:** The image shows a daytime driving scene under clear weather conditions. The road ahead is a two-lane road with a smooth surface, marked by white lane lines. There are no visible vehicles or pedestrians in the immediate vicinity. On the right side of the road, there is a construction area with orange traffic cones and a small excavator, indicating road work in progress. The road narrows slightly as it approaches the construction zone. There are no traffic lights visible in the image. The surrounding area is lined with trees, suggesting a suburban or semi-rural setting. The overall scene suggests that caution should be exercised as you approach the construction area, possibly reducing speed and being prepared to merge lanes if necessary.

**Vulnerable Road User (VRU) Identification:**
**Q:** Do you see any vulnerable road users within 20 meters ahead of you, such as cyclists, motorcyclists, or pedestrians?

**A:** No, I don't see any vulnerable road users ahead of me, such as bicycles, motorcycles, or pedestrians.

**Meta-action Planning:**
**Q:** Your current speed is 6 m/s, the navigation command is 'turn left', based on the understanding of the driving scene and the navigation information, what is your plan for the next three seconds? Please answer your SPEED plan and your PATH plan. SPEED includes KEEP, ACCELERATE and DECELERATE, and STOP, PATH includes STRAIGHT, RIGHT_CHANGE, LEFT_CHANGE, RIGHT_TURN, LEFT_TURN. For example, a correct answer format is like '<SPEED PATH PLAN>KEEP, LEFT_CHANGE'</SPEED PATH PLAN>.

**A:** <SPEED PATH PLAN>KEEP, RIGHT_TURN</SPEED PATH PLAN>

**Planning Explanation:**
**Q:** You are driving, your current speed is 5 m/s, and the navigation command is 'turn left', your driving decision for the next three seconds is to maintain the current speed and go straight. Based on the provided image of the driving environment, explain the most likely reason for this decision in one or two concise sentence.

**A:** The decision to maintain current speed and go straight is likely due to the presence of construction cones and a digger ahead, indicating ongoing road work that may require careful navigation and adherence to the current lane. The navigation command 'turn left' might be for a later point, as the immediate environment suggests caution rather than an immediate turn.

**End-to-End Trajectory Prediction:**
**Q:** You are an autonomous driving agent. You have access to multi-view camera images of a vehicle: (1) front view (which you should focus on with the most attention) <image>, (2) front right view <image>, and (3) front left view <image>. Your task is to do your best to predict future waypoints for the vehicle over the next 10 timesteps, given the vehicle's intent inferred from the images. Provided are the previous ego vehicle status recorded over the last 3.0 seconds (at 0.5-second intervals). This includes the x and y coordinates of the ego vehicle. Positive x means forward direction while positive y means leftwards. The data is presented in the format [x, y]:(t-3.0s) [-16.63, 1.76], Acceleration: X -0.06, Y -0.02 m/s^2, Velocity: X 5.41, Y 0.0 m/s, (t-2.5s) [-14.0, 1.27], Acceleration: X -0.07, Y -0.02 m/s^2, Velocity: X 5.34, Y 0.0 m/s, (t-2.0s) [-11.33, 0.85], Acceleration: X 0.17, Y 0.09 m/s^2, Velocity: X 5.32, Y 0.0 m/s, (t-1.5s) [-8.65, 0.49], Acceleration: X 0.27, Y 0.13 m/s^2, Velocity: X 5.37, Y 0.0 m/s, (t-1.0s) [-5.86, 0.21], Acceleration: X 0.48, Y 0.19 m/s^2, Velocity: X 5.5, Y 0.0 m/s, (t-0.5s) [-2.98, 0.05], Acceleration: X 0.64, Y 0.28 m/s^2, Velocity: X 5.72, Y 0.0 m/s, (t-0.0s) [0.0, 0.0], Acceleration: X 0.65, Y 0.17 m/s^2, Velocity: X 5.98, Y 0.0 m/s

**A:** <PLANNING>Predicted future movement details for the next 5 seconds (sampled at 0.5-second intervals), including BEV location in x and y directions (in meters). Positive x means forward direction while positive y means leftwards. The output is formatted as [x, y]: [-0.00, -0.00], [3.13, 0.05], [5.71, 0.11], [9.00, 0.22], [12.30, 0.33], [15.54, 0.43], [18.62, 0.49], [21.57, 0.55], [24.36, 0.60], [27.03, 0.67]</PLANNING>

**Traffic Signal Detection:**
**Q:** Given the provided forward-facing image <image> from a car's perspective, identify if there is a traffic light that affects the car's behavior. Respond with 'Red', 'Green', 'Yellow', or 'None'.

**A:** None

**Motion Intention Prediction:**
**Q:** You are driving, I will now provide you with the location and velocity information of dynamic objects in the front view image. Please predict their future driving behaviors, which can be divided into SPEED decisions and PATH decisions. SPEED includes KEEP, ACCELERATE, DECELERATE, and STOP, while PATH includes STRAIGHT, RIGHT_CHANGE, LEFT_CHANGE, RIGHT_TURN, and LEFT_TURN.I will now provide you with the position and velocity information of the dynamic objects:
Object 1: traffic_cone, 17 meters ahead, 2 meters to the right, speed of 0 m/s.
Object 2: traffic_cone, 12 meters ahead, 2 meters to the right, speed of 0 m/s.
Object 3: traffic_cone, 9 meters ahead, 4 meters to the right, speed of 0 m/s.
Object 4: traffic_cone, 11 meters ahead, 2 meters to the right, speed of 0 m/s.
Object 5: barrier, 28 meters ahead, 7 meters to the right, speed of 0 m/s.
Object 6: traffic_cone, 20 meters ahead, 1 meters to the right, speed of 0 m/s.
Object 7: traffic_cone, 25 meters ahead, 1 meters to the right, speed of 0 m/s.
Object 8: barrier, 25 meters ahead, 7 meters to the right, speed of 0 m/s.
Object 9: traffic_cone, 19 meters ahead, 1 meters to the right, speed of 0 m/s.
Object 10: traffic_cone, 13 meters ahead, 2 meters to the right, speed of 0 m/s.
Object 11: pedestrian, 29 meters ahead, 4 meters to the right, speed of 0 m/s.
Object 12: car, 29 meters ahead, 0 meters to the left, speed of 5 m/s.
Object 13: traffic_cone, 8 meters ahead, 3 meters to the right, speed of 0 m/s.
Object 14: traffic_cone, 14 meters ahead, 2 meters to the right, speed of 0 m/s.
Object 15: construction_vehicle, 28 meters ahead, 4 meters to the right, speed of 0 m/s.
Object 16: barrier, 23 meters ahead, 3 meters to the right, speed of 0 m/s.
Object 17: traffic_cone, 21 meters ahead, 1 meters to the right, speed of 0 m/s.
Object 18: traffic_cone, 22 meters ahead, 1 meters to the right, speed of 0 m/s.
Object 19: traffic_cone, 18 meters ahead, 2 meters to the right, speed of 0 m/s.
Object 20: barrier, 31 meters ahead, 1 meters to the right, speed of 0 m/s.
Object 21: traffic_cone, 15 meters ahead, 2 meters to the right, speed of 0 m/s.
Object 22: traffic_cone, 9 meters ahead, 2 meters to the right, speed of 0 m/s.
Please predict the future driving behaviors of these objects based on the front view image. For example, a well-formatted answer should be like:
Object 1: KEEP, STRAIGHT
Object 2: DECELERATE, RIGHT_TURN
Object 3: ACCELERATE, LEFT_CHANGE
 For example, a correct answer format is like '<DYNAMIC OBJECTS>Object 1: KEEP, STRAIGHT
Object 2: STOP, STRAIGHT
</DYNAMIC OBJECTS>.

**A:** <DYNAMIC OBJECTS>Object 1: STOP, STRAIGHT
Object 2: STOP, STRAIGHT
Object 3: STOP, STRAIGHT
Object 4: STOP, STRAIGHT
Object 5: STOP, STRAIGHT
Object 6: STOP, STRAIGHT
Object 7: STOP, STRAIGHT
Object 8: STOP, STRAIGHT
Object 9: STOP, STRAIGHT
Object 10: STOP, STRAIGHT
Object 11: STOP, STRAIGHT
Object 12: KEEP, STRAIGHT
Object 13: STOP, STRAIGHT
Object 14: STOP, STRAIGHT
Object 15: STOP, STRAIGHT
Object 16: STOP, STRAIGHT
Object 17: STOP, STRAIGHT
Object 18: STOP, STRAIGHT
Object 19: STOP, STRAIGHT
Object 20: STOP, STRAIGHT
Object 21: STOP, STRAIGHT
Object 22: STOP, STRAIGHT

Figure 8: Q&A example from the nuScenes dataset

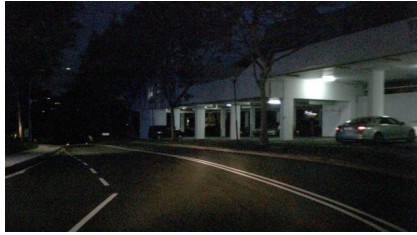

**Scene Description:**
**Q:** Suppose you are driving, and I'm providing you with the image captured by the car's front camera, generate a description of the driving scene which includes the key factors for driving planning, including the positions and movements of vehicles and pedestrians; prevailing weather conditions; time of day, distinguishing between daylight and nighttime; road conditions, indicating smooth surfaces or the presence of obstacles; and the status of traffic lights which influence your decision making, specifying whether they are red or green. The description should be concise, providing an accurate understanding of the driving environment to facilitate informed decision-making.

**A:** The image shows a nighttime driving scene with low ambient light, illuminated primarily by streetlights and the vehicle's headlights. The road appears to be a multi-lane highway with a smooth surface, no visible obstacles, and clear lane markings. There are no pedestrians in the scene.
A white car is positioned in the right lane ahead, moving in the same direction as the viewer's vehicle. The car is at a moderate distance, suggesting a steady flow of traffic. No other vehicles are immediately visible in the immediate lanes or in the oncoming traffic.
The weather conditions seem clear, with no signs of rain, fog, or snow, as the road surface is dry and the visibility is relatively good for nighttime conditions. The time of day is clearly nighttime, as indicated by the darkness of the sky and the artificial lighting.
There are no traffic lights visible in the image, so their status does not influence the immediate decision-making. However, the presence of the white car ahead suggests that the driver should maintain a safe following distance and be prepared to adjust speed if the car changes its behavior, such as slowing down or changing lanes. The overall environment appears to be conducive to safe driving, with no immediate hazards detected.

**Vulnerable Road User (VRU) Identification:**
**Q:** Do you see any vulnerable road users within 20 meters ahead of you, such as cyclists, motorcyclists, or pedestrians?

**A:** No, I don't see any vulnerable road users ahead of me, such as bicycles, motorcycles, or pedestrians.

**End-to-End Trajectory Prediction:**
**Q:** You are an autonomous driving agent. You have access to multi-view camera images of a vehicle: (1) front view (which you should focus on with the most attention) <image>, (2) front right view <image>, and (3) front left view <image>. Your task is to do your best to predict future waypoints for the vehicle over the next 10 timesteps, given the vehicle's intent inferred from the images. Provided are the previous ego vehicle status recorded over the last 3.0 seconds (at 0.5-second intervals). This includes the x and y coordinates of the ego vehicle. Positive x means forward direction while positive y means leftwards. The data is presented in the format [x, y]:(t-3.0s) [-20.54, 2.48], Acceleration: X -0.09, Y 0.12 m/s^2, Velocity: X 7.53, Y 0.0 m/s, (t-2.5s) [-16.77, 1.88], Acceleration: X -0.31, Y 0.15 m/s^2, Velocity: X 7.42, Y 0.0 m/s, (t-2.0s) [-13.47, 1.38], Acceleration: X -0.43, Y 0.11 m/s^2, Velocity: X 7.26, Y 0.0 m/s, (t-1.5s) [-9.91, 0.83], Acceleration: X -0.62, Y 0.17 m/s^2, Velocity: X 7.03, Y 0.0 m/s, (t-1.0s) [-6.49, 0.38], Acceleration: X -0.59, Y 0.63 m/s^2, Velocity: X 6.71, Y 0.0 m/s, (t-0.5s) [-3.18, 0.07], Acceleration: X -0.61, Y 0.76 m/s^2, Velocity: X 6.52, Y 0.0 m/s, (t-0.0s) [0.0, 0.0], Acceleration: X -0.25, Y 1.08 m/s^2, Velocity: X 6.39, Y 0.0 m/s

**A:** <PLANNING>Predicted future movement details for the next 5 seconds (sampled at 0.5-second intervals), including BEV location in x and y directions (in meters). Positive x means forward direction while positive y means leftwards. The output is formatted as [x, y]: [-0.00, 0.00], [3.17, 0.24], [6.37, 0.83], [9.55, 1.80], [12.94, 3.17], [15.70, 4.44], [18.10, 5.67], [21.12, 7.29], [24.07, 8.94], [27.14, 10.72]</PLANNING>

**Traffic Signal Detection:**
**Q:** Given the provided forward-facing image <image> from a car's perspective, identify if there is a traffic light that affects the car's behavior. Respond with 'Red', 'Green', 'Yellow', or 'None'.

**A:** None

**Motion Intention Prediction:**
**Q:** You are driving, I will now provide you with the location and velocity information of dynamic objects in the front view image. Please predict their future driving behaviors, which can be divided into SPEED decisions and PATH decisions. SPEED includes KEEP, ACCELERATE, DECELERATE, and STOP, while PATH includes STRAIGHT, RIGHT_CHANGE, LEFT_CHANGE, RIGHT_TURN, and LEFT_TURN.I will now provide you with the position and velocity information of the dynamic objects:
Object 1: car, 37 meters ahead, 9 meters to the right, speed of 0 m/s.
Object 2: car, 16 meters ahead, 7 meters to the right, speed of 0 m/s.
Please predict the future driving behaviors of these objects based on the front view image. For example, a well-formatted answer should be like:
Object 1: KEEP, STRAIGHT
Object 2: DECELERATE, RIGHT_TURN
Object 3: ACCELERATE, LEFT_CHANGE
For example, a correct answer format is like '<DYNAMIC OBJECTS>
Object 1: KEEP, STRAIGHT
Object 2: STOP, STRAIGHT
</DYNAMIC OBJECTS>'.

**A:** <DYNAMIC OBJECTS>
Object 1: KEEP, RIGHT_TURN
Object 2: STOP, STRAIGHT
Object 3: STOP, STRAIGHT
Object 4: STOP, STRAIGHT
Object 5: STOP, STRAIGHT
</DYNAMIC OBJECTS>

**Meta-action Planning:**
**Q:** Your current speed is 6 m/s, the navigation command is 'turn left', based on the understanding of the driving scene and the navigation information, what is your plan for the next three seconds? Please answer your SPEED plan and your PATH plan. SPEED includes KEEP, ACCELERATE and DECELERATE, and STOP, PATH includes STRAIGHT, RIGHT_CHANGE, LEFT_CHANGE, RIGHT_TURN, LEFT_TURN. For example, a correct answer format is like '<SPEED PATH PLAN>KEEP, LEFT_CHANGE</SPEED PATH PLAN>.

**A:** <SPEED PATH PLAN>KEEP, LEFT_TURN</SPEED PATH PLAN>

**Planning Explanation:**
**Q:** You are driving, your current speed is 6 m/s, and the navigation command is 'turn left', your driving decision for the next three seconds is to maintain the current speed and turn left. Based on the provided image of the driving environment, explain the most likely reason for this decision in one or two concise sentence.

**A:** The decision to maintain current speed and turn left is likely due to the presence of a visible left turn lane and the absence of oncoming traffic, ensuring a safe and smooth maneuver.

Figure 9: Q&A example from the nuScenes dataset

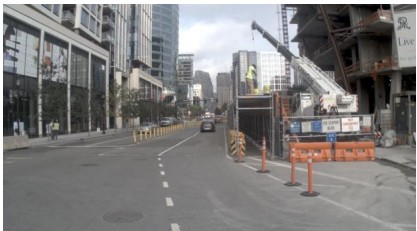

**Scene Description:**

**Q:** Suppose you are driving, and I'm providing you with the image captured by the car's front camera, generate a description of the driving scene which includes the key factors for driving planning, including the positions and movements of vehicles and pedestrians; prevailing weather conditions; time of day, distinguishing between daylight and nighttime; road conditions, indicating smooth surfaces or the presence of obstacles; and the status of traffic lights which influence your decision making, specifying whether they are red or green. The description should be concise, providing an accurate understanding of the driving environment to facilitate informed decision-making.

**A:** The scene is set during daylight hours under clear, sunny weather conditions, providing excellent visibility. The road surface appears smooth and well-maintained, with no visible obstacles. Traffic is moderate, with a few vehicles ahead in the same lane, all moving at a steady pace. There are no pedestrians visible in the immediate vicinity. The traffic light ahead is green, signaling that it is safe to proceed. Orange traffic cones are positioned along the right side of the road, indicating a possible construction zone or lane closure ahead, which may require a slight adjustment in driving path. The overall environment suggests a calm driving situation, with no immediate hazards, but drivers should remain cautious of the potential lane changes due to the construction area.

**Vulnerable Road User (VRU) Identification:**

**Q:** Do you see any vulnerable road users within 20 meters ahead of you, such as cyclists, motorcyclists, or pedestrians?

**A:** No, I don't see any vulnerable road users ahead of me, such as bicycles, motorcycles, or pedestrians.

**Meta-action Planning:**

**Q:** Your current speed is 5 m/s, the navigation command is go straight, based on the understanding of the driving scene and the navigation information, what is your plan for the next three seconds? Please answer your SPEED plan and your PATH plan. SPEED includes KEEP, ACCELERATE and DECELERATE, and STOP, PATH includes STRAIGHT, RIGHT_CHANGE, LEFT_CHANGE, RIGHT_TURN, LEFT_TURN. Based on the provided image of the driving environment, For example, a correct answer format is like '<SPEED PATH PLAN>KEEP, LEFT_CHANGE'</SPEED PATH PLAN>.

**A:** <SPEED PATH PLAN>KEEP, STRAIGHT</SPEED PATH PLAN>

**Traffic Signal Detection:**

**Q:** Given the provided forward-facing image <image> from a car's perspective, identify if there is a traffic light that affects the car's behavior. Respond with 'Red', 'Green', 'Yellow', or 'None'.

**A:** None

**Planning Explanation:**

**Q:** You are driving, your current speed is 5 m/s, and the navigation command is go straight your driving decision for the next three seconds is to maintain the current speed and go straight. Based on the provided image of the driving environment, explain the most likely reason for this decision in one or two concise sentence.

**A:** The decision to maintain the current speed and go straight is likely due to the clear road ahead with no immediate obstacles, and the presence of construction barriers and workers suggests a need for cautious, steady driving to ensure safety. The traffic light is green, allowing for uninterrupted forward movement.

**End-to-End Trajectory Prediction:**

**Q:** You are an autonomous driving agent. You have access to multi-view camera images of a vehicle: (1) front view (which you should focus on with the most attention) <image>, (2) front right view <image>, and (3) front left view <image>. Your task is to do your best to predict future waypoints for the vehicle over the next 10 timesteps, given the vehicle's intent inferred from the images. Provided are the previous ego vehicle statuses recorded over the last 1.5 seconds (at 0.5-second intervals). This includes the x and y coordinates of the ego vehicle. Positive x means forward direction while positive y means leftwards. The data is presented in the format [x, y]:(t-1.5s) [-6.17, -0.04], Acceleration: X 1.14, Y -0.16 m/s^2, Velocity: X 3.22, Y 0.09 m/s, (t-1.0s) [-4.4, -0.03], Acceleration: X 1.14, Y -0.1 m/s^2, Velocity: X 3.86, Y 0.08 m/s, (t-0.5s) [-2.34, -0.03], Acceleration: X 1.15, Y -0.13 m/s^2, Velocity: X 4.47, Y 0.13 m/s, (t-0.0s) [0.0, 0.0], Acceleration: X 1.03, Y 0.3 m/s^2, Velocity: X 5.01, Y 0.17 m/s

**A:** <PLANNING>Predicted future movement details for the next 5 seconds (sampled at 0.5-second intervals), including BEV location in x and y directions (in meters). Positive x means forward direction while positive y means leftwards. The output is formatted as [x, y]: [2.59, 0.07], [5.41, 0.22], [8.44, 0.43], [11.7, 0.65], [15.17, 0.82], [18.82, 0.9], [22.65, 0.87], [26.63, 0.77], [30.7, 0.64], [34.73, 0.49]</PLANNING>

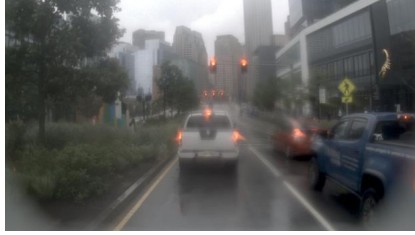

**Scene Description:**

**Q:** Suppose you are driving, and I'm providing you with the image captured by the car's front camera, generate a description of the driving scene which includes the key factors for driving planning, including the positions and movements of vehicles and pedestrians; prevailing weather conditions; time of day, distinguishing between daylight and nighttime; road conditions, indicating smooth surfaces or the presence of obstacles; and the status of traffic lights which influence your decision making, specifying whether they are red or green. The description should be concise, providing an accurate understanding of the driving environment to facilitate informed decision-making.

**A:** The scene is set during daylight hours under overcast weather conditions, with a light mist that slightly reduces visibility. The road surface appears wet, suggesting recent rain, and is marked with clear lane lines, indicating a smooth surface with no visible obstacles.
In the immediate lane ahead, a white sedan is traveling in the same direction, maintaining a steady pace and positioned centrally within the lane. To the right, a black SUV is also moving forward, slightly ahead of the sedan. There are no pedestrians visible in the scene.
The traffic light ahead is green, signaling that it is safe to proceed. However, due to the wet road conditions and reduced visibility, it is advisable to maintain a safe following distance from the vehicles ahead and drive at a moderate speed. The overall environment suggests a need for cautious driving, with attention to the potential for slippery conditions and the need to be prepared for any sudden changes in traffic flow.

**Meta-action Planning:**

**Q:** Your current speed is 5 m/s, the navigation command is go straight, based on the understanding of the driving scene and the navigation information, what is your plan for the next three seconds? Please answer your SPEED plan and your PATH plan. SPEED includes KEEP, ACCELERATE and DECELERATE, and STOP, PATH includes STRAIGHT, RIGHT_CHANGE, LEFT_CHANGE, RIGHT_TURN, LEFT_TURN. Based on the provided image of the driving environment, For example, a correct answer format is like '<SPEED PATH PLAN>KEEP, LEFT_CHANGE'</SPEED PATH PLAN>.

**A:** <SPEED PATH PLAN>KEEP,STRAIGHT</SPEED PATH PLAN>

**Traffic Signal Detection:**

**Q:** Given the provided forward-facing image <image> from a car's perspective, identify if there is a traffic light that affects the car's behavior. Respond with 'Red', 'Green', 'Yellow', or 'None'.

**A:** None

**Planning Explanation:**

**Q:** You are driving, your current speed is 5 m/s, and the navigation command is go straight your driving decision for the next three seconds is to maintain the current speed and go straight. Based on the provided image of the driving environment, explain the most likely reason for this decision in one or two concise sentence.

**A:** The decision to maintain the current speed and go straight is likely due to the clear road ahead with no immediate obstacles or traffic, ensuring a safe and efficient path forward. The wet road conditions also suggest maintaining a steady, controlled speed to avoid slipping or losing traction.

**End-to-End Trajectory Prediction:**

**Q:** You are an autonomous driving agent. You have access to multi-view camera images of a vehicle: (1) front view (which you should focus on with the most attention) <image>, (2) front right view <image>, and (3) front left view <image>. Your task is to do your best to predict future waypoints for the vehicle over the next 10 timesteps, given the vehicle's intent inferred from the images. Provided are the previous ego vehicle statuses recorded over the last 1.5 seconds (at 0.5-second intervals). This includes the x and y coordinates of the ego vehicle. Positive x means forward direction while positive y means leftwards. The data is presented in the format [x, y]:(t-1.5s) [-11.29, 0.04], Acceleration: X -1.31, Y 0.05 m/s^2, Velocity: X 8.88, Y 0.16 m/s, (t-1.0s) [-7.2, -0.0], Acceleration: X -1.36, Y 0.04 m/s^2, Velocity: X 8.16, Y 0.13 m/s, (t-0.5s) [-3.46, -0.01], Acceleration: X -1.19, Y 0.02 m/s^2, Velocity: X 7.54, Y 0.15 m/s, (t-0.0s) [0.0, 0.0], Acceleration: X -1.17, Y 0.12 m/s^2, Velocity: X 6.95, Y 0.16 m/s

**A:** <PLANNING>Predicted future movement details for the next 5 seconds (sampled at 0.5-second intervals), including BEV location in x and y directions (in meters). Positive x means forward direction while positive y means leftwards. The output is formatted as [x, y]: [3.18, 0.01], [6.1, 0.04], [8.78, 0.05], [11.24, 0.07], [13.5, 0.08], [15.55, 0.08], [17.4, 0.09], [19.04, 0.09], [20.49, 0.1], [21.69, 0.1]</PLANNING>

Figure 10: Q&A examples from the NAVSIM dataset

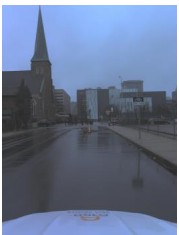

**Scene Description:**
**Q:** Suppose you are driving, and I'm providing you with the image captured by the car's front camera, generate a description of the driving scene which includes the key factors for driving planning, including the positions and movements of vehicles and pedestrians; prevailing weather conditions; time of day, distinguishing between daylight and nighttime; road conditions, indicating smooth surfaces or the presence of obstacles; and the status of traffic lights which influence your decision making, specifying whether they are red or green. The description should be concise, providing an accurate understanding of the driving environment to facilitate informed decision-making.

**A:** The scene is set during daylight hours under overcast weather conditions, with a wet road surface indicating recent or ongoing rain. The visibility is moderate, with no direct sunlight affecting the view. The road ahead is a multi-lane highway with a smooth surface, marked by clear lane dividers and a central median.
There are several vehicles ahead, including a black sedan directly in front of you, maintaining a steady speed in the right lane. A van is visible in the left lane, slightly ahead and to the left, also moving at a consistent pace. No pedestrians are present in the immediate vicinity of the road.
There are no traffic light in front of you. However, caution is advised due to the wet road conditions, which may reduce tire traction. No significant obstacles are present on the road, and the overall traffic flow appears to be light and orderly. The driver should maintain a safe following distance and be prepared for potential changes in road conditions or traffic flow.

**Vulnerable Road User (VRU) Identification:**
**Q:** Do you see any vulnerable road users within 20 meters ahead of you, such as cyclists, motorcyclists, or pedestrians?

**A:** No, I don't see any vulnerable road users ahead of me, such as bicycles, motorcycles, or pedestrians.

**Meta-action Planning:**
**Q:** Your current speed is 9 m/s, the navigation command is go straight, based on the understanding of the driving scene and the navigation information, what is your plan for the next three seconds? Please answer your SPEED plan and your PATH plan. SPEED includes KEEP, ACCELERATE and DECELERATE, and STOP, PATH includes STRAIGHT, RIGHT_CHANGE, LEFT_CHANGE, RIGHT_TURN, LEFT_TURN. Based on the provided image of the driving environment, For example, a correct answer format is like '<SPEED PATH PLAN>KEEP, LEFT_CHANGE'</SPEED PATH PLAN>.

**A:** <SPEED PATH PLAN>DECELERATE,STRAIGHT</SPEED PATH PLAN>

**Traffic Signal Detection:**
**Q:** Given the provided forward-facing image <image> from a car's perspective, identify if there is a traffic light that affects the car's behavior. Respond with 'Red', 'Green', 'Yellow', or 'None'.

**A:** None

**Planning Explanation:**
**Q:** You are driving, your current speed is 9 m/s, and the navigation command is go straight your driving decision for the next three seconds is to decelerate and go straight. Based on the provided image of the driving environment, explain the most likely reason for this decision in one or two concise sentence.

**A:** The decision to decelerate and go straight is likely due to the presence of a pedestrian crossing or a stop sign ahead, as indicated by the visible road markings and the urban environment. Additionally, the wet road conditions suggest a need for caution, further supporting the decision to slow down.

**End-to-End Trajectory Prediction:**
**Q:** You are an autonomous driving agent. You have access to multi-view camera images of a vehicle: (1) front view (which you should focus on with the most attention) <image>, (2) front right view <image>, and (3) front left view <image>. Your task is to do your best to predict future waypoints for the vehicle over the next 10 timesteps, given the vehicle's intent inferred from the images. Provided are the previous ego vehicle statuses recorded over the last 1.5 seconds (at 0.5-second intervals). This includes the x and y coordinates of the ego vehicle. Positive x means forward direction while positive y means leftwards. The data is presented in the format [x, y]:(t-1.5s) [-15.37, 0.24], Acceleration: X -0.06, Y -0.12 m/s^2, Velocity: X 10.37, Y -0.13 m/s, (t-1.0s) [-10.18, 0.18], Acceleration: X -0.64, Y 0.05 m/s^2, Velocity: X 10.34, Y -0.19 m/s, (t-0.5s) [-5.01, 0.08], Acceleration: X 0.00, Y 0.00 m/s^2, Velocity: X 10.02, Y -0.16 m/s, (t-0.0s) [0.0, 0.0], Acceleration: X -1.15, Y 0.82 m/s^2, Velocity: X 10.02, Y -0.16 m/s

**A:** <PLANNING>Predicted future movement details for the next 5 seconds (sampled at 0.5-second intervals), including BEV location in x and y directions (in meters). Positive x means forward direction while positive y means leftwards. The output is formatted as [x, y]: [5.01, -0.08], [9.73, 0.04], [13.97, 0.41], [17.76, 0.95], [21.08, 1.55], [24.04, 2.10], [26.66, 2.55], [29.04, 2.87], [31.15, 3.03], [32.82, 3.01]</PLANNING>

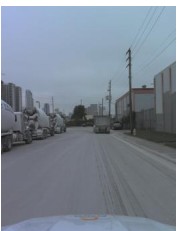

**Scene Description:**
**Q:** Suppose you are driving, and I'm providing you with the image captured by the car's front camera, generate a description of the driving scene which includes the key factors for driving planning, including the positions and movements of vehicles and pedestrians; prevailing weather conditions; time of day, distinguishing between daylight and nighttime; road conditions, indicating smooth surfaces or the presence of obstacles; and the status of traffic lights which influence your decision making, specifying whether they are red or green. The description should be concise, providing an accurate understanding of the driving environment to facilitate informed decision-making.

**A:** The scene is set during daylight hours under overcast skies, suggesting potential for rain but currently dry road conditions. The road surface appears smooth with no visible obstacles, and tire tracks indicate recent vehicle activity. Directly ahead, a line of vehicles, including a prominent white truck, is moving slowly, suggesting moderate traffic flow. No pedestrians are visible in the immediate vicinity. Traffic lights are not in view, but the slow movement of vehicles ahead implies a possible upcoming stop or a traffic light further down the road. The surrounding environment includes urban infrastructure with buildings and utility poles, indicating a city or suburban setting. The overall driving environment requires cautious navigation due to the presence of other vehicles and the potential for changing weather conditions.

**Meta-action Planning:**
**Q:** Your current speed is 4 m/s, the navigation command is go straight, based on the understanding of the driving scene and the navigation information, what is your plan for the next three seconds? Please answer your SPEED plan and your PATH plan. SPEED includes KEEP, ACCELERATE and DECELERATE, and STOP, PATH includes STRAIGHT, RIGHT_CHANGE, LEFT_CHANGE, RIGHT_TURN, LEFT_TURN. Based on the provided image of the driving environment, For example, a correct answer format is like '<SPEED PATH PLAN>KEEP, LEFT_CHANGE'</SPEED PATH PLAN>.

**A:** <SPEED PATH PLAN>KEEP,STRAIGHT</SPEED PATH PLAN>

**Planning Explanation:**
**Q:** You are driving, your current speed is 4 m/s, and the navigation command is go straight your driving decision for the next three seconds is to maintain the current speed and go straight. Based on the provided image of the driving environment, explain the most likely reason for this decision in one or two concise sentence.

**A:** The decision to maintain the current speed and go straight is likely due to the clear road ahead with no immediate obstacles or traffic, as indicated by the open space and the absence of vehicles or pedestrians in the provided views. The environment appears to be a construction or industrial area with minimal activity, further supporting the choice to continue straight at a steady pace.

**Traffic Signal Detection:**
**Q:** Given the provided forward-facing image <image> from a car's perspective, identify if there is a traffic light that affects the car's behavior. Respond with 'Red', 'Green', 'Yellow', or 'None'.

**A:** None

**Vulnerable Road User (VRU) Identification:**
**Q:** Do you see any vulnerable road users within 20 meters ahead of you, such as cyclists, motorcyclists, or pedestrians?

**A:** No, I don't see any vulnerable road users ahead of me, such as bicycles, motorcycles, or pedestrians.

**Motion Intention Prediction:**
**Q:** You are driving, I will now provide you with the location and relative velocity (to your vehicle) information of dynamic objects in the front center view image <image>. Please predict their future driving behaviors, which can be divided into SPEED decisions and PATH decisions. SPEED includes KEEP, ACCELERATE, DECELERATE, and STOP, while PATH includes STRAIGHT, RIGHT_CHANGE, LEFT_CHANGE, RIGHT_TURN, and LEFT_TURN. I will now provide you with the position and velocity information of the dynamic objects:
Object 1: TRUCK, 39 meters ahead, 2 meters to the right, speed of 4 m/s.
Please predict the future driving behaviors of these objects based on the front view image. For example, a well-formatted answer should be like '<DYNAMIC OBJECTS>
Object 1: KEEP, STRAIGHT
Object 2: DECELERATE, RIGHT_TURN
Object 3: ACCELERATE, LEFT_CHANGE
For example, a correct answer format is like '<DYNAMIC OBJECTS>
Object 1: KEEP, STRAIGHT
Object 2: STOP, STRAIGHT
</DYNAMIC OBJECTS>.

**A:** <DYNAMIC OBJECTS>Object 1: DECELERATE, STRAIGHT
</DYNAMIC OBJECTS>

Figure 11: Q&A examples from the Argoverse-V2 dataset

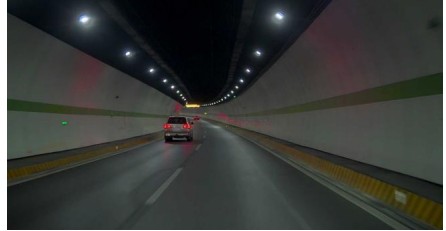

**Scene Description:**
**Q:** Suppose you are driving, and I'm providing you with the image captured by the car's front camera, generate a description of the driving scene which includes the key factors for driving planning, including the positions and movements of vehicles and pedestrians; prevailing weather conditions; time of day, distinguishing between daylight and nighttime; road conditions, indicating smooth surfaces or the presence of obstacles; and the status of traffic lights which influence your decision making, specifying whether they are red or green. The description should be concise, providing an accurate understanding of the driving environment to facilitate informed decision-making.

**A:** The scene is set inside a well-lit tunnel during nighttime, as indicated by the artificial lighting and the absence of natural light. The road surface appears smooth and clear, with no visible obstacles. The tunnel is equipped with overhead lights that provide consistent illumination throughout the passage.
There are two vehicles ahead, both moving in the same direction as the observer. The vehicle directly in front is a silver sedan, maintaining a steady pace in the right lane. Further ahead, a second vehicle is visible, also in the right lane, suggesting a consistent flow of traffic. No pedestrians are present in the scene.
The traffic lights at the tunnel's exit are not visible in the image, so their status cannot be determined. However, the green arrow on the tunnel wall suggests that the lane is designated for a specific direction, likely straight ahead or a slight turn. The overall environment is conducive to smooth driving, with no immediate hazards or interruptions. Drivers should maintain a safe following distance from the vehicles ahead and be prepared for any changes in traffic conditions as they exit the tunnel.

**Meta-action Planning:**
**Q:** Your current speed is 19 m/s, the navigation command is go straight, based on the understanding of the driving scene and the navigation information, what is your plan for the next three seconds? Please answer your SPEED plan and your PATH plan. SPEED includes KEEP, ACCELERATE and DECELERATE, and STOP, PATH includes STRAIGHT, RIGHT_CHANGE, LEFT_CHANGE, RIGHT_TURN, LEFT_TURN. Based on the provided image of the driving environment, For example, a correct answer format is like '<SPEED PATH PLAN>KEEP, LEFT_CHANGE'</SPEED PATH PLAN>.

**A:** <SPEED PATH PLAN>KEEP,STRAIGHT</SPEED PATH PLAN>

**Traffic Signal Detection:**
**Q:** Given the provided forward-facing image <image> from a car's perspective, identify if there is a traffic light that affects the car's behavior. Respond with 'Red', 'Green', 'Yellow', or 'None'.

**A:** None

**Planning Explanation:**
**Q:** You are driving, your current speed is 19 m/s, and the navigation command is go straight your driving decision for the next three seconds is to maintain the current speed and go straight. Based on the provided image of the driving environment, explain the most likely reason for this decision in one or two concise sentence.

**A:** The decision to maintain the current speed and go straight is likely due to the clear, unobstructed path ahead in the tunnel, with no immediate turns, obstacles, or other vehicles requiring a change in speed or direction.

**End-to-End Trajectory Prediction:**
**Q:** You are an autonomous driving agent. You have vehicle's front center image <image>. Your task is to do your best to predict future waypoints for the vehicle over the next 10 timesteps, given the vehicle's intent inferred from the images. Provided are the previous ego vehicle statuses recorded over the last 1.5 seconds (at 0.5-second intervals). This includes the x and y coordinates of the ego vehicle. Positive x means forward direction while positive y means leftwards. The data is presented in the format [x, y]:(t-1.5s) [-29.14, 0.62], Acceleration: X -0.66, Y 0.41 m/s^2, Velocity: X 19.47, Y -0.61 m/s, (t-1.0s) [-19.40, 0.31], Acceleration: X 1.02, Y 0.35 m/s^2, Velocity: X 19.14, Y -0.40 m/s, (t-0.5s) [-9.83, 0.11], Acceleration: X -38.86, Y 39.76 m/s^2, Velocity: X 19.65, Y -0.22 m/s, (t-0.0s) [0.0, 0.0], Acceleration: X -0.85, Y -39.86 m/s^2, Velocity: X 0.22, Y 19.65 m/s

**A:** <PLANNING>Predicted future movement details for the next 5 seconds (sampled at 0.5-second intervals), including BEV location in x and y directions (in meters). Positive x means forward direction while positive y means leftwards. The output is formatted as [x, y]: [9.83, 0.11], [19.52, 0.12], [29.06, 0.02], [38.78, -0.18], [48.32, -0.43], [57.90, -0.76], [67.51, -1.23], [77.01, -1.78], [86.37, -2.39], [95.66, -3.12]</PLANNING>

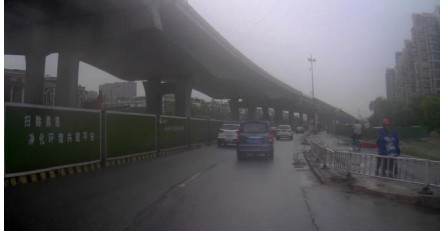

**Scene Description:**
**Q:** Suppose you are driving, and I'm providing you with the image captured by the car's front camera, generate a description of the driving scene which includes the key factors for driving planning, including the positions and movements of vehicles and pedestrians; prevailing weather conditions; time of day, distinguishing between daylight and nighttime; road conditions, indicating smooth surfaces or the presence of obstacles; and the status of traffic lights which influence your decision making, specifying whether they are red or green. The description should be concise, providing an accurate understanding of the driving environment to facilitate informed decision-making.

**A:** The driving scene is set during daylight hours under overcast weather conditions, with no direct sunlight and a slightly damp road surface, suggesting recent rain. The road is a multi-lane urban street with a smooth surface, bordered by a green construction barrier on the right side. The barrier has Chinese characters printed on it, indicating a construction or maintenance zone ahead.
There are several vehicles ahead, including a blue car directly in front, maintaining a steady pace. The vehicles are spaced at a safe distance, with no immediate signs of aggressive driving or sudden stops. On the right side of the road, near the construction barrier, a pedestrian wearing a blue shirt and a red helmet is walking along the sidewalk, moving away from the camera's perspective.
The traffic light ahead is not visible in the image, but the flow of traffic suggests it is likely green or the vehicles are moving under a permissive signal. There are no visible obstacles on the road, and the overall environment appears to be calm and orderly, conducive to smooth driving. However, the presence of the construction zone and the pedestrian requires extra caution and attention to potential changes in road conditions or unexpected movements.

**Meta-action Planning:**
**Q:** Your current speed is 8 m/s, the navigation command is go straight, based on the understanding of the driving scene and the navigation information, what is your plan for the next three seconds? Please answer your SPEED plan and your PATH plan. SPEED includes KEEP, ACCELERATE and DECELERATE, and STOP, PATH includes STRAIGHT, RIGHT_CHANGE, LEFT_CHANGE, RIGHT_TURN, LEFT_TURN. Based on the provided image of the driving environment, For example, a correct answer format is like '<SPEED PATH PLAN>KEEP, LEFT_CHANGE'</SPEED PATH PLAN>.

**A:** <SPEED PATH PLAN>KEEP,STRAIGHT</SPEED PATH PLAN>

**Traffic Signal Detection:**
**Q:** Given the provided forward-facing image <image> from a car's perspective, identify if there is a traffic light that affects the car's behavior. Respond with 'Red', 'Green', 'Yellow', or 'None'.

**A:** None

**Planning Explanation:**
**Q:** You are driving, your current speed is 8 m/s, and the navigation command is go straight your driving decision for the next three seconds is to maintain the current speed and go straight. Based on the provided image of the driving environment, explain the most likely reason for this decision in one or two concise sentence.

**A:** The decision to maintain the current speed and go straight is likely due to the clear road ahead with no immediate obstacles or traffic, and the presence of construction barriers on the right side, which suggests a need to stay in the current lane for safety.

**End-to-End Trajectory Prediction:**
**Q:** You are an autonomous driving agent. You have vehicle's front center image <image>. Your task is to do your best to predict future waypoints for the vehicle over the next 10 timesteps, given the vehicle's intent inferred from the images. Provided are the previous ego vehicle statuses recorded over the last 1.5 seconds (at 0.5-second intervals). This includes the x and y coordinates of the ego vehicle. Positive x means forward direction while positive y means leftwards. The data is presented in the format [x, y]:(t-1.5s) [-12.59, 0.34], Acceleration: X -0.07, Y 0.17 m/s^2, Velocity: X 8.42, Y -0.31 m/s, (t-1.0s) [-8.38, 0.18], Acceleration: X -0.01, Y 0.15 m/s^2, Velocity: X 8.39, Y -0.22 m/s, (t-0.5s) [-4.19, 0.07], Acceleration: X -16.47, Y 17.06 m/s^2, Velocity: X 8.38, Y -0.15 m/s, (t-0.0s) [0.0, 0.0], Acceleration: X -0.35, Y -16.77 m/s^2, Velocity: X 0.15, Y 8.38 m/s

**A:** <PLANNING>Predicted future movement details for the next 5 seconds (sampled at 0.5-second intervals), including BEV location in x and y directions (in meters). Positive x means forward direction while positive y means leftwards. The output is formatted as [x, y]: [4.19, 0.07], [8.38, 0.13], [12.58, 0.32], [16.82, 0.71], [21.10, 1.24], [25.42, 1.90], [29.79, 2.63], [34.19, 3.35], [38.54, 4.07], [42.86, 4.85]</PLANNING>

Figure 12: Q&A examples from the ONCE dataset

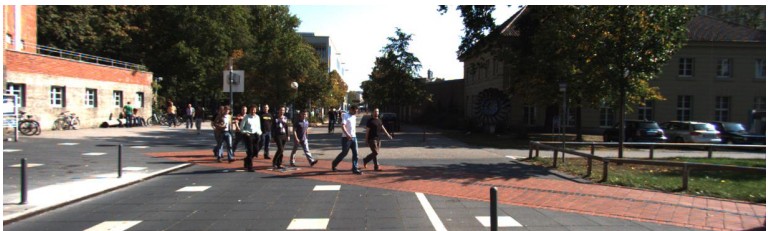

**Scene Description:**
**Q:** Suppose you are driving, and I'm providing you with the image captured by the car's front camera, generate a description of the driving scene which includes the key factors for driving planning, including the positions and movements of vehicles and pedestrians; prevailing weather conditions; time of day, distinguishing between daylight and nighttime; road conditions, indicating smooth surfaces or the presence of obstacles; and the status of traffic lights which influence your decision making, specifying whether they are red or green. The description should be concise, providing an accurate understanding of the driving environment to facilitate informed decision-making.

**A:** The scene is set during daylight hours under clear, sunny weather conditions, which provide excellent visibility. The road surface appears smooth and free of any visible obstacles, ensuring a clear path for driving. A group of pedestrians is crossing the road at a designated crosswalk, moving from left to right in the image. They are positioned in the middle of the crosswalk, indicating that they have already started crossing and are in the process of completing it. On the right side of the image, a line of vehicles is visible, including a white car and a black car, both waiting behind the crosswalk. These vehicles are stationary, likely due to the pedestrians crossing. The traffic light for the direction the vehicles are facing is red, which explains their halted state and allows the pedestrians to cross safely.
For driving planning, it is crucial to maintain a complete stop until the pedestrians have fully crossed the road. The current conditions suggest a need for patience and adherence to traffic rules to ensure the safety of all road users. The clear weather and smooth road conditions are favorable, but the presence of pedestrians are the key factors influencing the immediate driving decision.

**Meta-action Planning:**
**Q:** Your current speed is 0 m/s, the navigation command is go straight, based on the understanding of the driving scene and the navigation information, what is your plan for the next three seconds? Please answer your SPEED plan and your PATH plan. SPEED includes KEEP, ACCELERATE and DECELERATE, and STOP, PATH includes STRAIGHT, RIGHT_CHANGE, LEFT_CHANGE, RIGHT_TURN, LEFT_TURN. Based on the provided image of the driving environment, For example, a correct answer format is like '<SPEED PATH PLAN>KEEP, LEFT_CHANGE'</SPEED PATH PLAN>.

**A:** <SPEED PATH PLAN>STOP,STRAIGHT</SPEED PATH PLAN>

**Traffic Signal Detection:**
**Q:** Given the provided forward-facing image <image> from a car's perspective, identify if there is a traffic light that affects the car's behavior. Respond with 'Red', 'Green', 'Yellow', or 'None'.
**A:** None

**Planning Explanation:**
**Q:** You are driving, your current speed is 0 m/s, and the navigation command is go straight your driving decision for the next three seconds is to stop the car. Based on the provided image of the driving environment, explain the most likely reason for this decision in one or two concise sentence.

**A:** The decision to stop the car is likely due to the presence of pedestrians crossing the road ahead, ensuring their safety before proceeding.

**End-to-End Trajectory Prediction:**
**Q:** You are an autonomous driving agent. You have access to front camera image of a vehicle: <image>. Your task is to do your best to predict future waypoints for the vehicle over the next 10 timesteps, given the vehicle's intent inferred from the images. Provided are the previous ego vehicle statuses recorded over the last 1.5 seconds (at 0.5-second intervals). This includes the x and y coordinates of the ego vehicle. Positive x means forward direction while positive y means leftwards. The data is presented in the format [x, y]:(t-1.5s) [0.01, -0.01], Acceleration: X 0.01, Y 0.00 m/s^2, Velocity: X -0.01, Y 0.00 m/s, (t-1.0s) [0.01, -0.01], Acceleration: X -0.00, Y 0.01 m/s^2, Velocity: X -0.00, Y 0.00 m/s, (t-0.5s) [0.00, -0.00], Acceleration: X 0.00, Y 0.00 m/s^2, Velocity: X -0.01, Y 0.01 m/s, (t-0.0s) [0.0, 0.0], Acceleration: X -0.01, Y -0.00 m/s^2, Velocity: X -0.01, Y 0.01 m/s

**A:** <PLANNING>Predicted future movement details for the next 5 seconds (sampled at 0.5-second intervals), including BEV location in x and y directions (in meters). Positive x means forward direction while positive y means leftwards. The output is formatted as [x, y]: [-0.00, 0.00], [-0.01, 0.01], [-0.01, 0.01], [-0.01, 0.01], [-0.01, 0.00], [-0.01, 0.00], [-0.00, -0.00], [-0.00, -0.01], [-0.00, -0.02], [-0.00, -0.02]</PLANNING>

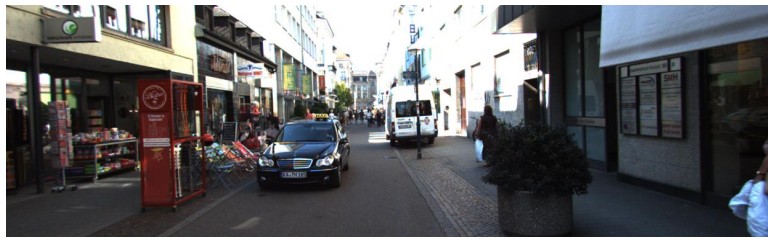

**Scene Description:**
**Q:** Suppose you are driving, and I'm providing you with the image captured by the car's front camera, generate a description of the driving scene which includes the key factors for driving planning, including the positions and movements of vehicles and pedestrians; prevailing weather conditions; time of day, distinguishing between daylight and nighttime; road conditions, indicating smooth surfaces or the presence of obstacles; and the status of traffic lights which influence your decision making, specifying whether they are red or green. The description should be concise, providing an accurate understanding of the driving environment to facilitate informed decision-making.

**A:** The scene is set during daylight hours under clear, sunny weather conditions, which provide excellent visibility. The road surface appears smooth and free of any significant obstacles, ensuring a clear path for driving. On the left side of the road, a white van is parked, and a blue car is moving ahead in the same direction as the driver, maintaining a safe distance. On the right side, a pedestrian is walking along the sidewalk, moving away from the driver's direction, and there are no immediate signs of crossing intentions.
There are no traffic lights in the immediate view, but the driver should remain cautious and attentive to the pedestrian's movements and the blue car's speed, as well as any other potential changes in the environment. The overall driving environment is relatively calm and predictable, which should facilitate smooth and safe driving.

**Meta-action Planning:**
**Q:** Your current speed is 3 m/s, the navigation command is go straight, based on the understanding of the driving scene and the navigation information, what is your plan for the next three seconds? Please answer your SPEED plan and your PATH plan. SPEED includes KEEP, ACCELERATE and DECELERATE, and STOP, PATH includes STRAIGHT, RIGHT_CHANGE, LEFT_CHANGE, RIGHT_TURN, LEFT_TURN. Based on the provided image of the driving environment, For example, a correct answer format is like '<SPEED PATH PLAN>KEEP, LEFT_CHANGE'</SPEED PATH PLAN>.

**A:** <SPEED PATH PLAN>KEEP,STRAIGHT</SPEED PATH PLAN>

**Traffic Signal Detection:**
**Q:** Given the provided forward-facing image <image> from a car's perspective, identify if there is a traffic light that affects the car's behavior. Respond with 'Red', 'Green', 'Yellow', or 'None'.

**A:** None

**Planning Explanation:**
**Q:** You are driving, your current speed is 3 m/s, and the navigation command is go straight your driving decision for the next three seconds is to maintain the current speed and go straight. Based on the provided image of the driving environment, explain the most likely reason for this decision in one or two concise sentence.

**A:** The decision to maintain the current speed and go straight is likely due to the clear, unobstructed road ahead and the absence of any immediate traffic or pedestrians, ensuring a safe and smooth continuation of the journey.

**End-to-End Trajectory Prediction:**
**Q:** You are an autonomous driving agent. You have access to front camera image of a vehicle: <image>. Your task is to do your best to predict future waypoints for the vehicle over the next 10 timesteps, given the vehicle's intent inferred from the images. Provided are the previous ego vehicle statuses recorded over the last 1.5 seconds (at 0.5-second intervals). This includes the x and y coordinates of the ego vehicle. Positive x means forward direction while positive y means leftwards. The data is presented in the format [x, y]:(t-1.5s) [-5.28, -0.29], Acceleration: X 0.19, Y -0.20 m/s^2, Velocity: X 3.44, Y 0.28 m/s, (t-1.0s) [-3.56, -0.14], Acceleration: X 0.13, Y -0.15 m/s^2, Velocity: X 3.53, Y 0.18 m/s, (t-0.5s) [-1.80, -0.05], Acceleration: X 0.00, Y 0.00 m/s^2, Velocity: X 3.60, Y 0.11 m/s, (t-0.0s) [0.0, 0.0], Acceleration: X -0.13, Y -0.20 m/s^2, Velocity: X 3.60, Y 0.11 m/s

**A:** <PLANNING>Predicted future movement details for the next 5 seconds (sampled at 0.5-second intervals), including BEV location in x and y directions (in meters). Positive x means forward direction while positive y means leftwards. The output is formatted as [x, y]: [1.80, 0.05], [3.56, 0.06], [5.04, 0.07], [6.06, 0.08], [6.57, 0.10], [7.00, 0.11], [7.48, 0.12], [8.10, 0.14], [8.96, 0.19], [10.11, 0.29]</PLANNING>

Figure 13: Q&A examples from the KITTI dataset

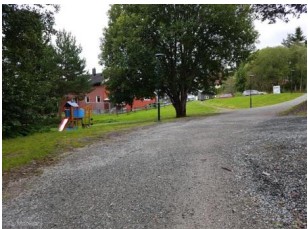

**Scene Description:**
**Q:** Suppose you are driving, and I'm providing you with the image captured by the car's front camera, generate a description of the driving scene which includes the key factors for driving planning, including the positions and movements of vehicles and pedestrians; prevailing weather conditions; time of day, distinguishing between daylight and nighttime; road conditions, indicating smooth surfaces or the presence of obstacles; and the status of traffic lights which influence your decision making, specifying whether they are red or green. The description should be concise, providing an accurate understanding of the driving environment to facilitate informed decision-making.

**A:** The scene is set during daylight hours under overcast skies, suggesting diffused natural light with no direct sunlight, which implies good visibility without glare. The road is a gravel surface, appearing dry and free of significant obstacles, though the uneven texture requires cautious driving to avoid potential jolts or loss of traction. There are no visible vehicles or pedestrians in the immediate vicinity, indicating a clear path ahead. The area is bordered by lush greenery and a playground with a slide and swing set, suggesting a residential or park area where children might be present, necessitating vigilance. No traffic lights are visible in the image, so decision-making is based on the absence of other traffic and the need to maintain a safe speed in a potentially pedestrian-heavy zone. The overall environment is calm, with no immediate hazards, but the presence of a playground calls for extra caution.

**Meta-action Planning:**
**Q:** Your current speed is 4 m/s, the navigation command is go straight and turn right, based on the understanding of the driving scene and the navigation information, what is your plan for the next three seconds? Please answer your SPEED plan and your PATH plan. SPEED includes KEEP, ACCELERATE and DECELERATE, and STOP, PATH includes STRAIGHT, RIGHT_CHANGE, LEFT_CHANGE, RIGHT_TURN, LEFT_TURN. Based on the provided image of the driving environment, For example, a correct answer format is like '<SPEED PATH PLAN>KEEP, LEFT_CHANGE'</SPEED PATH PLAN>.

**A:** <SPEED PATH PLAN>KEEP,STRAIGHT</SPEED PATH PLAN>

**Traffic Signal Detection:**
**Q:** Given the provided forward-facing image <image> from a car's perspective, identify if there is a traffic light that affects the car's behavior. Respond with 'Red', 'Green', 'Yellow', or 'None'.

**A:** None

**Planning Explanation:**
**Q:** You are driving, your current speed is 4 m/s, and the navigation command is go straight and turn right your driving decision for the next three seconds is to maintain the current speed and go straight. Based on the provided image of the driving environment, explain the most likely reason for this decision in one or two concise sentence.

**A:** The decision to maintain the current speed and go straight is likely due to the clear, unobstructed path ahead and the absence of immediate turns or obstacles, ensuring a safe and smooth continuation towards the upcoming right turn.

**End-to-End Trajectory Prediction:**
**Q:** You are an autonomous driving agent. You have access to front camera image of a vehicle: <image>. Your task is to do your best to predict future waypoints for the vehicle over the next 10 timesteps, given the vehicle's intent inferred from the images. Provided are the previous ego vehicle statuses recorded over the last 1.5 seconds (at 0.5-second intervals). This includes the x and y coordinates of the ego vehicle. Positive x means forward direction while positive y means leftwards. The data is presented in the format [x, y]:(t-1.5s) [-7.46, -0.72], Acceleration: X -4.89, Y -1.09 m/s^2, Velocity: X 6.54, Y 0.93 m/s, (t-1.0s) [-4.19, -0.25], Acceleration: X 0.39, Y -0.51 m/s^2, Velocity: X 4.10, Y 0.38 m/s, (t-0.5s) [-2.14, -0.06], Acceleration: X 0.00, Y 0.00 m/s^2, Velocity: X 4.29, Y 0.13 m/s, (t-0.0s) [0.0, 0.0], Acceleration: X -0.56, Y -0.29 m/s^2, Velocity: X 4.29, Y 0.13 m/s

**A:** <PLANNING>Predicted future movement details for the next 5 seconds (sampled at 0.5-second intervals), including BEV location in x and y directions (in meters). Positive x means forward direction while positive y means leftwards. The output is formatted as [x, y]: [2.14, 0.06], [4.15, 0.05], [6.03, -0.05], [8.10, -0.29], [10.24, -0.67], [12.21, -1.27], [13.80, -2.16], [15.74, -3.44], [17.63, -4.79], [20.28, -6.83]</PLANNING>

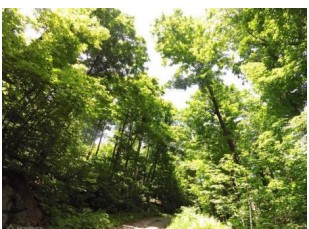

**Scene Description:**
**Q:** Suppose you are driving, and I'm providing you with the image captured by the car's front camera, generate a description of the driving scene which includes the key factors for driving planning, including the positions and movements of vehicles and pedestrians; prevailing weather conditions; time of day, distinguishing between daylight and nighttime; road conditions, indicating smooth surfaces or the presence of obstacles; and the status of traffic lights which influence your decision making, specifying whether they are red or green. The description should be concise, providing an accurate understanding of the driving environment to facilitate informed decision-making.

**A:** The image depicts a serene, forested area with a narrow, unpaved path winding through dense greenery. There are no vehicles or pedestrians visible in the scene, indicating a quiet and isolated environment. The weather appears to be clear and sunny, with bright daylight illuminating the scene, suggesting it is daytime. The road conditions are rough and natural, with no signs of asphalt or pavement, and the path is surrounded by vegetation, which could pose as obstacles for off-road vehicles. There are no traffic lights present in this natural setting, so decisions would be based on the natural terrain and the absence of other road users. The overall environment suggests a need for cautious and attentive driving, particularly mindful of the uneven and potentially obstructed path.

**Meta-action Planning:**
**Q:** Your current speed is 16 m/s, the navigation command is go straight and turn right, based on the understanding of the driving scene and the navigation information, what is your plan for the next three seconds? Please answer your SPEED plan and your PATH plan. SPEED includes KEEP, ACCELERATE and DECELERATE, and STOP, PATH includes STRAIGHT, RIGHT_CHANGE, LEFT_CHANGE, RIGHT_TURN, LEFT_TURN. Based on the provided image of the driving environment, For example, a correct answer format is like '<SPEED PATH PLAN>KEEP, LEFT_CHANGE'</SPEED PATH PLAN>.

**A:** <SPEED PATH PLAN>DECELERATE,STRAIGHT</SPEED PATH PLAN>

**Vulnerable Road User (VRU) Identification:**
**Q:** Do you see any vulnerable road users within 20 meters ahead of you, such as cyclists, motorcyclists, or pedestrians?

**A:** No, I don't see any vulnerable road users ahead of me, such as bicycles, motorcycles, or pedestrians.

**Planning Explanation:**
**Q:** You are driving, your current speed is 16 m/s, and the navigation command is go straight and turn right your driving decision for the next three seconds is to decelerate and go straight. Based on the provided image of the driving environment, explain the most likely reason for this decision in one or two concise sentence.

**A:** The decision to decelerate and go straight is likely due to the narrow, winding path surrounded by dense trees, which requires careful navigation to avoid obstacles and maintain control. The upcoming right turn may also necessitate a slower approach to safely execute the maneuver.

**End-to-End Trajectory Prediction:**
**Q:** You are an autonomous driving agent. You have access to front camera image of a vehicle: <image>. Your task is to do your best to predict future waypoints for the vehicle over the next 10 timesteps, given the vehicle's intent inferred from the images. Provided are the previous ego vehicle statuses recorded over the last 1.5 seconds (at 0.5-second intervals). This includes the x and y coordinates of the ego vehicle. Positive x means forward direction while positive y means leftwards. The data is presented in the format [x, y]:(t-1.5s) [-17.98, -6.51], Acceleration: X 1.31, Y 5.02 m/s^2, Velocity: X 8.97, Y 2.59 m/s, (t-1.0s) [-13.49, -5.21], Acceleration: X 15.46, Y 0.44 m/s^2, Velocity: X 9.63, Y 5.10 m/s, (t-0.5s) [-8.68, -2.66], Acceleration: X 0.00, Y 0.00 m/s^2, Velocity: X 17.36, Y 5.32 m/s, (t-0.0s) [0.0, 0.0], Acceleration: X -3.23, Y -17.07 m/s^2, Velocity: X 17.36, Y 5.32 m/s

**A:** <PLANNING>Predicted future movement details for the next 5 seconds (sampled at 0.5-second intervals), including BEV location in x and y directions (in meters). Positive x means forward direction while positive y means leftwards. The output is formatted as [x, y]: [8.68, 2.66], [16.55, 1.05], [23.88, -0.20], [26.57, -0.90], [29.72, -1.05], [33.30, -1.99], [37.04, -4.76], [38.96, -8.40], [41.02, -12.75], [47.46, -19.82]</PLANNING>

Figure 14: Q&A examples from the Mapillary dataset

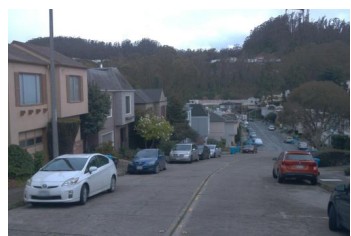

**Scene Description:**
**Q:** Suppose you are driving, and I'm providing you with the image captured by the car's front camera, generate a description of the driving scene which includes the key factors for driving planning, including the positions and movements of vehicles and pedestrians; prevailing weather conditions; time of day, distinguishing between daylight and nighttime; road conditions, indicating smooth surfaces or the presence of obstacles; and the status of traffic lights which influence your decision making, specifying whether they are red or green. The description should be concise, providing an accurate understanding of the driving environment to facilitate informed decision-making.

**A:** The scene is set during daylight hours under overcast skies, suggesting diffused natural light with no direct sunlight, which implies moderate visibility. The road appears to be a two-lane residential street with a smooth surface, free of visible obstacles. On the left side of the road, there are parked cars, including a white sedan and a blue sedan, indicating a no-parking zone or a residential parking area. A red car is seen further down the road, also parked on the left side. There are no pedestrians visible in the scene.
There is no traffic light ahead, the driver should remain cautious and be prepared to stop if any unexpected movements from other vehicles or pedestrians occur. The overall driving environment appears calm and controlled, with no immediate hazards or complications. The driver can proceed forward, keeping a safe distance from the parked cars and being mindful of the traffic light status.

**Vulnerable Road User (VRU) Identification:**
**Q:** Do you see any vulnerable road users within 20 meters ahead of you, such as cyclists, motorcyclists, or pedestrians?

**A:** No, I don't see any vulnerable road users ahead of me, such as bicycles, motorcycles, or pedestrians.

**End-to-End Trajectory Prediction:**
**Q:** You are an autonomous driving agent. You have access to multi-view camera images of a vehicle: (1) front view (which you should focus on with the most attention) <image>, (2) front right view <image>, and (3) front left view <image>. Your task is to do your best to predict future waypoints for the vehicle over the next 10 timesteps, given the vehicle's intent inferred from the images. Provided are the previous ego vehicle statuses recorded over the last 1.5 seconds (at 0.5-second intervals). This includes the x and y coordinates of the ego vehicle. Positive x means forward direction while positive y means leftwards. The data is presented in the format [x, y]:(t-1.5s) [-7.06, -1.25], Acceleration: X 0.48, Y -1.28 m/s^2, Velocity: X 4.40, Y 1.44 m/s, (t-1.0s) [-4.86, -0.53], Acceleration: X 0.88, Y -1.08 m/s^2, Velocity: X 4.64, Y 0.80 m/s, (t-0.5s) [-2.54, -0.13], Acceleration: X 1.28, Y -0.92 m/s^2, Velocity: X 5.08, Y 0.26 m/s, (t-0.0s) [0.0, 0.0], Acceleration: X 1.20, Y -1.04 m/s^2, Velocity: X 5.72, Y -0.20 m/s

**A:** <PLANNING>Predicted future movement details for the next 5 seconds (sampled at 0.5-second intervals), including BEV location in x and y directions (in meters). Positive x means forward direction while positive y means leftwards. The output is formatted as [x, y]: [2.86, -0.10], [6.02, -0.46], [9.34, -1.01], [12.92, -1.67], [16.76, -2.42], [20.84, -3.36], [25.05, -4.48], [29.34, -5.78], [33.71, -7.24], [38.12, -8.84]</PLANNING>

**Traffic Signal Detection:**
**Q:** Given the provided forward-facing image <image> from a car's perspective, identify if there is a traffic light that affects the car's behavior. Respond with 'Red', 'Green', 'Yellow', or 'None'.

**A:** None

**Motion Intention Prediction:**
**Q:** You are driving, I will now provide you with the location and velocity information of dynamic objects in the front center view image <image>. Please predict their future driving behaviors, which can be divided into SPEED decisions and PATH decisions. SPEED includes KEEP, ACCELERATE, DECELERATE, and STOP, while PATH includes STRAIGHT, RIGHT_CHANGE, LEFT_CHANGE, RIGHT_TURN, and LEFT_TURN.I will now provide you with the position and velocity information of the dynamic objects:
Object 1: Vehicle, 32 meters ahead, 1 meters to the left, speed of 0 m/s.
For example, a correct answer format is like '<DYNAMIC OBJECTS>
Object 1: KEEP, STRAIGHT
Object 2: STOP, STRAIGHT</DYNAMIC OBJECTS>.

**A:** <DYNAMIC OBJECTS>
Object 1: STOP, STRAIGHT
</DYNAMIC OBJECTS>

**Meta-action Planning:**
**Q:** Your current speed is 5 m/s, the navigation command is go straight, based on the understanding of the driving scene and the navigation information, what is your plan for the next three seconds? Please answer your SPEED plan and your PATH plan. SPEED includes KEEP, ACCELERATE and DECELERATE, and STOP, PATH includes STRAIGHT, RIGHT_CHANGE, LEFT_CHANGE, RIGHT_TURN, LEFT_TURN. Based on the provided image of the driving environment, For example, a correct answer format is like '<SPEED PATH PLAN>KEEP, LEFT_CHANGE'</SPEED PATH PLAN>.

**A:** <SPEED PATH PLAN>KEEP,RIGHT_TURN</SPEED PATH PLAN>

**Planning Explanation:**
**Q:** You are driving, your current speed is 5 m/s, and the navigation command is go straight your driving decision for the next three seconds is to maintain the current speed and turn right. Based on the provided image of the driving environment, explain the most likely reason for this decision in one or two concise sentence.

**A:** The decision to maintain speed and then turn right is likely due to the presence of a parked car on the right side of the road, indicating an upcoming need to navigate around it or follow a road that curves to the right. The clear visibility of the road ahead and the absence of immediate obstacles support maintaining the current speed until the turn is necessary.

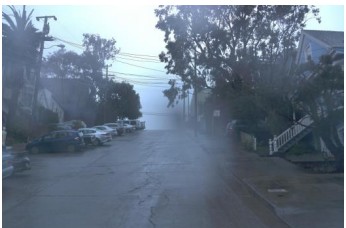

**Scene Description:**
**Q:** Suppose you are driving, and I'm providing you with the image captured by the car's front camera, generate a description of the driving scene which includes the key factors for driving planning, including the positions and movements of vehicles and pedestrians; prevailing weather conditions; time of day, distinguishing between daylight and nighttime; road conditions, indicating smooth surfaces or the presence of obstacles; and the status of traffic lights which influence your decision making, specifying whether they are red or green. The description should be concise, providing an accurate understanding of the driving environment to facilitate informed decision-making.

**A:** The scene is set during daylight hours under overcast weather conditions, with a light mist that slightly reduces visibility. The road surface appears smooth and free of significant obstacles, though the mist may obscure smaller hazards. There are no visible pedestrians in the immediate vicinity.
On the left side of the road, a line of parked cars is visible, with no immediate signs of movement. The traffic ahead is sparse, with a few vehicles visible in the distance, traveling in the same direction. These vehicles are maintaining a steady pace, suggesting no immediate traffic congestion.
There are no traffic lights visible in the current field of view, so decisions should be based on the flow of traffic and road signs. The road is flanked by trees and houses, which are partially obscured by the mist. The overall environment suggests a suburban or residential area.
Given the conditions, it is advisable to maintain a moderate speed, stay alert for any sudden changes in traffic conditions, and be prepared for reduced visibility due to the mist. The absence of traffic lights means paying extra attention to road signs and the behavior of other vehicles.

**Vulnerable Road User (VRU) Identification:**
**Q:** Do you see any vulnerable road users within 20 meters ahead of you, such as cyclists, motorcyclists, or pedestrians?

**A:** No, I don't see any vulnerable road users ahead of me, such as bicycles, motorcycles, or pedestrians.

**Meta-action Planning:**
**Q:** Your current speed is 3 m/s, the navigation command is turn right, based on the understanding of the driving scene and the navigation information, what is your plan for the next three seconds? Please answer your SPEED plan and your PATH plan. SPEED includes KEEP, ACCELERATE and DECELERATE, and STOP, PATH includes STRAIGHT, RIGHT_CHANGE, LEFT_CHANGE, RIGHT_TURN, LEFT_TURN. Based on the provided image of the driving environment, For example, a correct answer format is like '<SPEED PATH PLAN>KEEP, LEFT_CHANGE'</SPEED PATH PLAN>.

**A:** <SPEED PATH PLAN>KEEP,STRAIGHT</SPEED PATH PLAN>

**Traffic Signal Detection:**
**Q:** Given the provided forward-facing image <image> from a car's perspective, identify if there is a traffic light that affects the car's behavior. Respond with 'Red', 'Green', 'Yellow', or 'None'.

**A:** None

**Planning Explanation:**
**Q:** You are driving, your current speed is 5 m/s, and the navigation command is go straight your driving decision for the next three seconds is to maintain the current speed and change to the left lane. Based on the provided image of the driving environment, explain the most likely reason for this decision in one or two concise sentence.

**A:** The decision to maintain the current speed and change to the left lane is likely due to the presence of a parked truck in the right lane, which could obstruct the path if the vehicle ahead slows down or stops. Additionally, the left lane appears to be clearer, providing a safer and more efficient route forward.

**Motion Intention Prediction:**
**Q:** You are driving, I will now provide you with the location and velocity information of dynamic objects in the front center view image <image>. Please predict their future driving behaviors, which can be divided into SPEED decisions and PATH decisions. SPEED includes KEEP, ACCELERATE, DECELERATE, and STOP, while PATH includes STRAIGHT, RIGHT_CHANGE, LEFT_CHANGE, RIGHT_TURN, and LEFT_TURN.I will now provide you with the position and velocity information of the dynamic objects:
Object 1: Vehicle, 36 meters ahead, 8 meters to the left, speed of 0 m/s.
For example, a correct answer format is like '<DYNAMIC OBJECTS>Object 1: KEEP, STRAIGHT
Object 2: STOP, STRAIGHT
</DYNAMIC OBJECTS>.

**A:** <DYNAMIC OBJECTS>Object 1: STOP, STRAIGHT
</DYNAMIC OBJECTS>

Figure 15: Q&A examples from the Waymo dataset

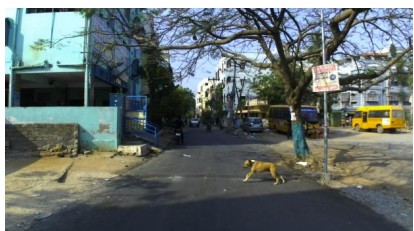

**Scene Description:**
**Q:** Suppose you are driving, and I'm providing you with the image captured by the car's front camera, generate a description of the driving scene which includes the key factors for driving planning, including the positions and movements of vehicles and pedestrians; prevailing weather conditions; time of day, distinguishing between daylight and nighttime; road conditions, indicating smooth surfaces or the presence of obstacles; and the status of traffic lights which influence your decision making, specifying whether they are red or green. The description should be concise, providing an accurate understanding of the driving environment to facilitate informed decision-making.

**A:** The scene is set during daylight hours under clear, sunny weather conditions, which provide excellent visibility. The road surface appears smooth and free of major obstacles, though a stray dog is crossing the road ahead, moving from left to right. This requires immediate attention and a potential reduction in speed to avoid an accident. A yellow auto-rickshaw is visible further ahead on the right side of the road, moving in the same direction as the vehicle. There are no pedestrians immediately near the vehicle, but a few are visible further down the road, walking along the sidewalk. The road is lined with trees and buildings, and the overall environment suggests an urban or suburban setting. The driver should be cautious of the crossing dog and maintain a safe distance from the auto-rickshaw while continuing to monitor the road for any additional hazards, and the driver must remain vigilant and prepared to stop if necessary.

**Meta-action Planning:**
**Q:** Your current speed is 3 m/s, the navigation command is go straight, based on the understanding of the driving scene and the navigation information, what is your plan for the next three seconds? Please answer your SPEED plan and your PATH plan. SPEED includes KEEP, ACCELERATE and DECELERATE, and STOP, PATH includes STRAIGHT, RIGHT_CHANGE, LEFT_CHANGE, RIGHT_TURN, LEFT_TURN. Based on the provided image of the driving environment, For example, a correct answer format is like '<SPEED PATH PLAN>KEEP, LEFT_CHANGE'</SPEED PATH PLAN>.

**A:** <SPEED PATH PLAN>KEEP,STRAIGHT</SPEED PATH PLAN>

**Traffic Signal Detection:**
**Q:** Given the provided forward-facing image <image> from a car's perspective, identify if there is a traffic light that affects the car's behavior. Respond with 'Red', 'Green', 'Yellow', or 'None'.

**A:** None

**Planning Explanation:**
**Q:** You are driving, your current speed is 3 m/s, and the navigation command is go straight your driving decision for the next three seconds is to maintain the current speed and go straight. Based on the provided image of the driving environment, explain the most likely reason for this decision in one or two concise sentence.

**A:** The decision to maintain the current speed and go straight is likely due to the clear road ahead with no immediate obstacles, and the presence of a dog crossing the road necessitates cautious, steady driving to ensure safety.

**End-to-End Trajectory Prediction:**
**Q:** You are an autonomous driving agent. You have access to front camera image of a vehicle: <image>. Your task is to do your best to predict future waypoints for the vehicle over the next 10 timesteps, given the vehicle's intent inferred from the images. Provided are the previous ego vehicle statuses recorded over the last 1.5 seconds (at 0.5-second intervals). This includes the x and y coordinates of the ego vehicle. Positive x means forward direction while positive y means leftwards. The data is presented in the format [x, y]:(t-1.5s) [-8.93, -0.04], Acceleration: X -1.68, Y -0.12 m/s^2, Velocity: X 6.86, Y 0.07 m/s, (t-1.0s) [-5.50, -0.01], Acceleration: X -2.12, Y -0.02 m/s^2, Velocity: X 6.02, Y 0.01 m/s, (t-0.5s) [-2.48, 0.00], Acceleration: X 0.00, Y 0.00 m/s^2, Velocity: X 4.97, Y -0.00 m/s, (t-0.0s) [0.0, 0.0], Acceleration: X -2.32, Y -0.09 m/s^2, Velocity: X 4.97, Y -0.00 m/s

**A:** <PLANNING>Predicted future movement details for the next 5 seconds (sampled at 0.5-second intervals), including BEV location in x and y directions (in meters). Positive x means forward direction while positive y means leftwards. The output is formatted as [x, y]: [2.48, -0.00], [4.39, -0.02], [5.74, -0.02], [6.82, -0.07], [7.76, -0.16], [8.67, -0.27], [9.42, -0.33], [10.32, -0.40], [11.45, -0.50], [13.02, -0.63]</PLANNING>

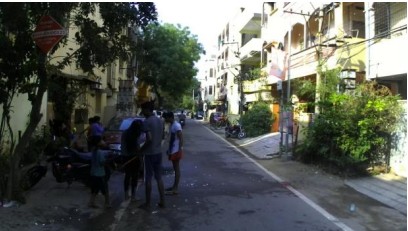

**Scene Description:**
**Q:** Suppose you are driving, and I'm providing you with the image captured by the car's front camera, generate a description of the driving scene which includes the key factors for driving planning, including the positions and movements of vehicles and pedestrians; prevailing weather conditions; time of day, distinguishing between daylight and nighttime; road conditions, indicating smooth surfaces or the presence of obstacles; and the status of traffic lights which influence your decision making, specifying whether they are red or green. The description should be concise, providing an accurate understanding of the driving environment to facilitate informed decision-making.

**A:** The scene is set during daylight hours under clear weather conditions, with ample sunlight enhancing visibility. The road surface appears smooth and free of significant obstacles, though there are a few scattered leaves and minor debris. A group of pedestrians is gathered on the left side of the road, near the sidewalk, seemingly engaged in conversation and not actively crossing. A motorcycle is parked on the left side, close to the pedestrians, and a car is visible further down the road, moving away from the camera's perspective. There are no traffic lights in the immediate view. The road is lined with trees and buildings, suggesting an urban or suburban setting. The overall environment is calm, with no immediate signs of heavy traffic or urgent pedestrian movement. Drivers should prepare to stop, maintaining a safe distance from the pedestrians and the parked motorcycle. The clear weather and good visibility conditions are favorable for driving, but attention should be given to the pedestrians and the stationary vehicles.

**Meta-action Planning:**
**Q:** Your current speed is 1 m/s, the navigation command is go straight, based on the understanding of the driving scene and the navigation information, what is your plan for the next three seconds? Please answer your SPEED plan and your PATH plan. SPEED includes KEEP, ACCELERATE and DECELERATE, and STOP, PATH includes STRAIGHT, RIGHT_CHANGE, LEFT_CHANGE, RIGHT_TURN, LEFT_TURN. Based on the provided image of the driving environment, For example, a correct answer format is like '<SPEED PATH PLAN>KEEP, LEFT_CHANGE'</SPEED PATH PLAN>.

**A:** <SPEED PATH PLAN>STOP,STRAIGHT</SPEED PATH PLAN>

**Traffic Signal Detection:**
**Q:** Given the provided forward-facing image <image> from a car's perspective, identify if there is a traffic light that affects the car's behavior. Respond with 'Red', 'Green', 'Yellow', or 'None'.

**A:** None

**Planning Explanation:**
**Q:** You are driving, your current speed is 1 m/s, and the navigation command is go straight your driving decision for the next three seconds is to stop the car. Based on the provided image of the driving environment, explain the most likely reason for this decision in one or two concise sentence.

**A:** The decision to stop the car is likely due to the presence of pedestrians crossing the road and a motorcycle parked on the side, which could pose a safety hazard if the vehicle continues moving. Additionally, the narrow street and parked vehicles on both sides may limit maneuverability, necessitating a stop.

**End-to-End Trajectory Prediction:**
**Q:** You are an autonomous driving agent. You have access to front camera image of a vehicle: <image>. Your task is to do your best to predict future waypoints for the vehicle over the next 10 timesteps, given the vehicle's intent inferred from the images. Provided are the previous ego vehicle statuses recorded over the last 1.5 seconds (at 0.5-second intervals). This includes the x and y coordinates of the ego vehicle. Positive x means forward direction while positive y means leftwards. The data is presented in the format [x, y]:(t-1.5s) [-2.18, -0.18], Acceleration: X 0.37, Y -0.25 m/s^2, Velocity: X 1.29, Y 0.24 m/s, (t-1.0s) [-1.53, -0.06], Acceleration: X 0.24, Y -0.23 m/s^2, Velocity: X 1.47, Y 0.12 m/s, (t-0.5s) [-0.80, 0.00], Acceleration: X 0.00, Y 0.00 m/s^2, Velocity: X 1.59, Y -0.00 m/s, (t-0.0s) [0.0, 0.0], Acceleration: X 0.51, Y -0.21 m/s^2, Velocity: X 1.59, Y -0.00 m/s

**A:** <PLANNING>Predicted future movement details for the next 5 seconds (sampled at 0.5-second intervals), including BEV location in x and y directions (in meters). Positive x means forward direction while positive y means leftwards. The output is formatted as [x, y]: [0.80, -0.00], [1.72, -0.05], [2.85, -0.15], [4.12, -0.23], [5.55, -0.22], [7.14, -0.12], [8.88, 0.04], [10.82, 0.36], [13.06, 0.76], [15.60, 1.25]</PLANNING>

Figure 16: Q&A examples from the IDD dataset

