# OpenReview forum: "Impromptu VLA: Open Weights and Open Data for Driving Vision-Language-Action Models"
_NeurIPS.cc/2025/Datasets_and_Benchmarks_Track — NeurIPS 2025 Datasets and Benchmarks Track poster_

### Official Review · Reviewer_6FzN · 2025-06-25

**Rating:** 4
**Confidence:** 3

**Summary:**

Considering the data scarcity for autonomous driving in unstructured environments, this paper introduces the Impromptu VLA Dataset, which contains 80k clips featuring rich multi-task QA annotations and corresponding action trajectories. Experiments with VLMs trained on this dataset achieve significant performance gains on closed-loop safety and open-loop trajectory prediction. The Impromptu VLA Dataset offers a valuable new resource to handle diverse and challenging unstructured road scenarios in perception, prediction, and planning capabilities.

**Dataset Code Accessibility:**

Yes

**Ethical Considerations:**

No, there are no or only very minor ethics concerns

**Final Justification:**

My concerns have been addressed mostly. I choose to keep my positive score.

**Limitations Weaknesses:**

1.	For Figure2(a), comparison of the proposed Impromptu VLA dataset with previous datasets is recommended for comprehensive illustration.
2.	Datasets are collected based on existing datasets. Targeted for unstructured environments, more specific selection and design are required for unfamiliar scenarios.
3.	The boost of performance seems to contribute to the scale and the richness of the dataset, combined with VLM finetuning. More ablations or verifications on the scale or types of data are required to expose the fundamental issues.

**Strengths Contributions:**

1.	This paper is well-written and easy-to-follow. Figures are clear to express the design and ideas.
2.	The dataset is comprehensive and worth investigation for real-world application in the field of autonomous driving.
3.	The crucial issues of unstructured scenarios are pointed out, and targeted solutions and annotations are proposed for this problem.
4.	Comprehensive experimental evaluations are conducted on closed-loop and open-loop benchmarks with the proposed dataset

---

> ### Author Rebuttal · Authors · 2025-07-30
>
> We've carefully considered the reviewer's valuable feedback and have prepared our responses to the points raised.
>
> ---
>
> 1. **Dataset Comparison**
>
> Actually, we have mentioned the quantity of original data as well as the amount of data of Impromptu-VLA in Table 1. However, we appreciate your suggestion to visualize this data directly in Figure 2(a) to more intuitively and comprehensively demonstrate the scale of our dataset relative to its underlying data source, and it will be included in the final version.
>
> 2. **Design for Unfamiliar Scenarios**
>
> You also rightly point out that datasets collected based on existing sources, particularly when targeting unstructured environments and unfamiliar scenarios, require more specific selection and design. We fully agree with this perspective. Achieving truly intelligent driving capabilities is an iterative process, not an overnight endeavor. The development of autonomous driving, as a field, has evolved from mastering foundational capabilities (e.g., simple straight-line driving, left/right turns) in more straightforward scenes. Building upon this, we systematically expand to cover more long-tail scenario distributions, which is precisely where our Impromptu VLA dataset makes its contribution by curating data specifically for these unstructured environments. We anticipate future work will propose richer and more comprehensive datasets to expand this distribution and propel autonomous driving forward, but our current work represents a significant step.
>
> Furthermore, we place considerable trust in the emergent capabilities of advanced models. We hypothesize that by training on our curated data from various unstructured scenarios, the model will develop the potential to generalize effectively to novel unstructured situations it has not explicitly encountered during training. Ultimately, a primary contribution of this paper is to define and formalize the challenges posed by these unstructured scenarios, rather than to offer a complete, one-size-fits-all solution. We eagerly anticipate and welcome subsequent research that will build upon this foundation and further advance the field.
>
> 3. **Ablation Studies**
>
> For your suggestion on ablations or verifications regarding data scale or types, we conducted a series of **seven ablation experiments**, systematically removing different categories of QA data during training and evaluating their impact. The results are presented in the table below:
>
> | QA data types not used for training | V.R.U | T.Light | Dyn.Obj. | M.P. | L2 (m) 1s | L2 (m) 2s | L2 (m) 3s | L2 (m) 4s | L2 (m) Avg. |
> | :---------------------------------- | :---- | :------ | :------- | :--- | :-------- | :-------- | :-------- | :-------- | :---------- |
> | V.R.U | 0.91 | 0.96 | 0.92 | 0.83 | 0.13 | 0.37 | 0.74 | 1.23 | 1.83 |
> | Dyn. Obj. | 0.92 | 0.96 | **0.66** | 0.83 | 0.13 | 0.38 | 0.76 | 1.25 | 1.86 |
> | Plan Exp. | 0.92 | 0.96 | 0.92 | 0.84 | 0.13 | 0.38 | 0.75 | 1.24 | 1.84 |
> | T. Light | 0.92 | 0.96 | 0.92 | 0.84 | 0.11 | 0.35 | 0.71 | 1.19 | 1.78 |
> | Scene Des. | 0.92 | 0.96 | 0.92 | 0.84 | 0.11 | 0.36 | 0.73 | 1.22 | 1.83 |
> | M.P. | 0.92 | 0.96 | 0.92 | **0.66** | 0.11 | 0.34 | 0.70 | 1.17 | 1.76 |
> | Traj. Pred. | 0.92 | 0.96 | 0.92 | 0.84 | 3.37 | 5.72 | 7.99 | 10.54 | 12.97 |
>
> Most metrics remain relatively close, but we observed two interesting data points: when **Dyn. Obj.** or **M.P.** QA data were excluded from training, the corresponding metrics (Dyn. Obj. accuracy was 0.20, M.P. accuracy was 0.56 for the Base Model) were still higher than the **Base Model's** performance. This suggests that **other QA data categories contribute positively to the model's performance on these specific tasks**, even when their direct supervision is removed. This highlights that our dataset serves as a robust diagnostic tool, effectively revealing the model's learned capabilities and the complex interplay between different types of driving knowledge. For example, improved scene understanding derived from training on one QA type might indirectly benefit other perception-based tasks. We emphasize that our primary contribution lies in proposing this novel dataset, filling a critical gap in the field.

---

### Official Review · Reviewer_HYe3 · 2025-06-28

**Rating:** 4
**Confidence:** 4

**Summary:**

This paper proposed a new dataset aiming to help VLA/VLMs address unstructured/corner case driving scenarios. It is curated from 8 open-sourced datasets, with meticulous post-processing to align their format, like frequency, and label each scenario with more textual information by using LLM. After human verification, the released version includes 80K clips with categories chosen from 4 pre-defined ones and contextual information labeled to provide surrounding information like vulnerable road users.

The closed-loop and open-loop experiments on the common nuScenes and NeuroNCAP validation benchmark show that the base VLM gains performance improvements by including the proposed data in training. In particular, the proposed dataset can improve the base VLM by a large margin when evaluated on the held-out test split of the proposed dataset in terms of the object detection/planning ability, showing its effectiveness for helping address unstructured driving scenarios.

The dataset and the post-processing tools will be made public.

**Additional Feedback:**

**Typo:**

1. Caption of Fig.2: each category. Notably

**Dataset Code Accessibility:**

Yes

**Dataset Code Comments:**

The code and data are well structured and ready to use.

**Ethical Considerations:**

No, there are no or only very minor ethics concerns

**Final Justification:**

After reading the author's response and other reviews, I tend to accept this paper.

**Limitations Weaknesses:**

Some benchmark results of other methods are needed for Table 4. For methods listed in *Ours and Key Competitors (Specialized Driving Models)*, it would be better to benchmark them on the test split of the proposed dataset. Even though fine-tuning each of them with the proposed dataset is impossible, we can at least do zero-shot benchmarking. Maybe some of them can do a good job in zero-shot as they are trained with more data before, while the base VLM is only trained with the proposed dataset. The authors can also fine-tune some of them to see if there are improvements to support the effectiveness of the proposed data. This can also save a lot of work for future users who are interested in using this dataset/benchmark. In addition, the baseline *3B-Base* is not fine-tuned with any driving data here, so the comparison is unfair. I suggest including at least the result of *3B-nuScenes* into this table. It should be easy as the *3B-nuScens* ckpt exists already.

**Strengths Contributions:**

1. The ability to solve corner cases is important in self-driving, where interaction with the physical world is needed, while data and benchmarks are missing for the VLM-based planning. Thus, this paper bridges this gap with both data and data processing tools open-sourced.

2. The experimental results indeed show the effectiveness of including this data in the training. It shows better results on traditional benchmarks (nuScenes) and the held-out test split of the proposed dataset.

3. The paper writing is comprehensive, providing thorough information for learning about this field.

---

> ### Author Rebuttal · Authors · 2025-07-30
>
> We've carefully considered the reviewer's valuable feedback and have prepared our responses to the points raised.
>
> ---
>
> ### Benchmarking Other Methods and Baseline Inclusion
>
> We understand the reviewer's suggestion to include benchmark results from other specialized driving models (DriveVLM, OmniDrive, EMMA) in Table 4, either through zero-shot evaluation or by fine-tuning them with our proposed dataset. We also agree on the importance of including the **3B-nuScenes** baseline.
>
> Regarding **DriveVLM and EMMA**, we couldn't benchmark them as they're not open-source, which inherently limits external evaluation and reproducibility.
>
> Regarding OmniDrive, although open source, we encountered significant challenges that prevented us from reliably benchmarking it on our dataset. Notably, OmniDrive proposes two architectures, Omni-L and Omni-Q, and we cite results from Omni-Q++ in our paper. However, we found that the OmniDrive public repository lacks the pre-trained weights for Omni-Q and Omni-Q++ described in the paper. The weights provided in the repository are only for 2D LLM initialization and the visual encoder/projection module. These require training with OmniDrive data to produce Omni-Q or Omni-Q++. However, the repository does not specify the configuration used to train the different models, making accurate reproduction difficult. Furthermore, while the trained ".pth" weight files were later released, the specific model (Omni-Q, Omni-Q++, or Omni-L) was not specified. Other issues also occurred, including a failure to converge during training and near-zero mAP values during validation. We have also observed other users reporting multiple similar issues on GitHub, even when strictly following the official instructions.
>
> Given these extensive issues—including missing training procedures, unclear model information, frequent runtime errors, severe environment conflicts, and community-reported irreproducibility—we can't guarantee that any reproduced results would align with the original paper. We emphasize these challenges to ensure transparency and uphold the principle of experimental reproducibility. The difficulty in reproducing existing methods highlights a key problem our work aims to address: providing an open-source, readily reproducible framework to foster advancements in this domain.
>
> ---
>
> ### Inclusion of 3B-nuScenes Baseline
>
> We agree with the reviewer's suggestion to include the **3B-Base+nuScenes** results. We've conducted the experiment, and the results are presented in the table below:
>
> | Method/QA | V.R.U | T.Light | Dyn.Obj. | M.P. | L2 (m) 1s | L2 (m) 2s | L2 (m) 3s | L2 (m) 4s | L2 (m) Avg. |
> | :---------------- | :----- | :----- | :----- | :----- | :-------- | :-------- | :-------- | :-------- | :---------- |
> | **3B Base** | 0.87 | 0.95 | 0.20 | 0.56 | 3.39 | 5.31 | 7.70 | 10.08 | 6.62 |
> | **3B Base+Impromptu** | **0.91** | **0.96** | **0.92** | **0.84** | **0.16** | **0.43** | **0.82** | **1.34** | **0.69** |
> | **3B Base+nuScenes** | 0.90 | **0.96** | 0.75 | 0.72 | 1.12 | 1.62 | 2.24 | 3.00 | 2.00 |
>
> As shown, **3B Base+nuScenes** generally performs better than **3B Base**, with the **L2 Loss (Avg. at 4s) reducing from 6.62 to 2.00**. This indicates that training on nuScenes data indeed helps the model acquire end-to-end trajectory prediction capabilities.
>
> However, the performance of **3B Base+nuScenes** falls short of **3B Base+Impromptu** in most metrics (excluding V.R.U and T.Light, where the base model already performs well). The **L2 Loss (Avg. at 4s) for 3B Base+nuScenes is 2.00**, notably higher than 0.69 for **3B Base+Impromptu**. This gap highlights the **diversity and richness of our Impromptu dataset**. As depicted in Figure 2(b) of our paper, nuScenes data tends to be more concentrated, while our dataset features a wider variety of trajectories. This broader diversity enables models trained on Impromptu to generalize better to more varied driving scenarios. The improvements in other tasks further suggest that training with end-to-end trajectory prediction data enhances the model's overall scene understanding, leading to better performance across different QA tasks. This implies a potential underlying connection between these diverse QA tasks, where improved scene understanding from trajectory prediction benefits other downstream capabilities.
>
> ---
>
> ### Typographical Correction
> We appreciate the reviewer pointing out the typo in the caption of Fig. 2. We'll fix this in the final version of the paper.

---

### Official Review · Reviewer_jfjU · 2025-07-01

**Rating:** 5
**Confidence:** 3

**Summary:**

This paper presents Impromptu VLA, a large-scale, richly annotated dataset aimed at improving Vision-Language-Action (VLA) models for autonomous driving, particularly in unstructured and rare scenarios. The dataset comprises 80K curated video clips from 8 open-source driving datasets and is organized into four taxonomy-defined unstructured categories: (1) roads with unclear boundaries, (2) temporary traffic rule changes, (3) unconventional dynamic obstacles, and (4) challenging road conditions.
The authors introduce a novel data curation pipeline using Chain-of-Thought (CoT) prompting with Qwen2.5-VL and verify outputs through human annotations. Empirical results show significant improvements in both closed-loop metrics and open-loop trajectory prediction on nuScenes when models are pre-trained on this dataset.

**Dataset Code Accessibility:**

Yes

**Dataset Code Comments:**

The dataset is available on huggingface.

**Ethical Considerations:**

No, there are no or only very minor ethics concerns

**Final Justification:**

I lean to accept this paper due to their foundational contributions to VLA data for autonomous driving.

**Limitations Weaknesses:**

1. Although the authors acknowledge that reliance on Qwen2.5-VL may induce bias in annotation, further quantitative evaluation of inter-annotator agreement or comparisons across different VLMs would strengthen the claim that bias is minimal.
2. The taxonomy seems reasonable, but it would be helpful to include ablation experiments showing the effect of removing specific categories or using simpler prompting strategies (e.g., flat captioning vs. CoT).

**Strengths Contributions:**

1. Impromptu VLA fills a critical gap in existing driving datasets by focusing specifically on unstructured environments, a domain underrepresented in current benchmarks.
2. The four-category taxonomy is well motivated and clearly defined, grounded in semantic clustering and VLM-based analysis.
3. Human-in-the-loop validation ensures label reliability despite using generative VLMs as part of the pipeline.
4. Experiments show significant improvements in both closed-loop metrics and open-loop trajectory prediction when models are pre-trained on this dataset.
5. Figures and tables are clear and informative, particularly the comparison plots showing qualitative improvements in decision-making under difficult conditions.

---

> ### Author Rebuttal · Authors · 2025-07-31
>
> We've carefully considered the reviewer's valuable feedback and have prepared our responses to the points raised.
>
> 1.  **Bias Induction from Qwen2.5-VL and Inter-annotator Agreement:**
>     We agree that relying on Qwen2.5-VL for initial annotation may introduce some bias. To mitigate this, we implemented a human verification step to cross-validate and refine the annotations. While a comprehensive quantitative evaluation of inter-annotator agreement or comparisons across various VLMs would indeed strengthen our claims, we faced significant cost constraints. Tagging the entire dataset, for instance, requires approximately 2000 GPU hours, making extensive comparisons across different VLMs computationally infeasible within the scope of this project.
>
> 2.  **Ablation Experiments for Taxonomy and Prompting Strategies:**
>     We appreciate your suggestion to include ablation experiments. We conducted a comparison of our CoT (Chain-of-Thought) prompting strategy against a simpler flat captioning approach on a subset of the Mapillary dataset, comprising 1000 images. The results demonstrate that our CoT strategy indeed improves classification accuracy.
>
>     Specifically, the precision for flat captioning was **0.63**, while our CoT approach achieved a precision of **0.70**. This indicates that the more structured reasoning facilitated by CoT leads to more accurate categorization.
>
>     The prompts used for this comparison are as follows:
>
>     **Flat Captioning Prompt:**
>
>     ```
>     You are an autonomous driving agent. You have access to the vehicle's front view image <image>. Please describe the overall scene, including the time of day (morning/noon/afternoon/night), weather conditions (sunny/cloudy/rainy), and road surface (muddy/gravel/snow/ice/no obvious road surface). Integrate details about visible roadway features such as lane lines, sidewalks, guardrails, signal lights, and traffic signs, explaining their direct impact and constraints on the vehicle's driving behavior. Concurrently, identify any movable objects like pedestrians, animals, or other vehicles, detailing their relative position to the autonomous vehicle and their potential influence on its path. Finally, provide a comprehensive judgment of the vehicle's environment by classifying it into one or more of the following categories, providing a clear justification for each: 1. Roads with unclear boundaries (ambiguous or undefined traversable paths, faded/absent markings); 2. Temporary traffic rule changes (construction zones, human traffic controllers, temporary signage); 3. Unconventional dynamic obstacles (large/erratically moving vehicles, vulnerable road users in unexpected locations, animal encounters); 4. Challenging road conditions (adverse surfaces like potholes, mud, snow, ice, or environmental conditions like fog, heavy rain, low-light, glare). If no matching categories are found, classify it as a standard road scenario. Ensure the entire analysis is consistent and avoids contradictions. Your final output MUST be a JSON object with a 'description' (this entire scene analysis) and a 'categories' key. The 'categories' value MUST be a list containing the names of the matched categories, or an empty list if no categories apply.
>     Your answer should be in the following format
>     {"description": "your description", "categories": []}
>     ```
>
>     **CoT Prompt:**
>
>     ```
>     You are an autonomous driving agent. You have access to the vehicle's front view image <image>. Your task is divided into the following steps:
>     The first step is to give a description of the scene from each viewpoint, then analyze the scene and give the corresponding time period and weather conditions, whether it is muddy, gravel, snow, ice, or a scene with no obvious road surface.
>     The second step is to analyze roadway features, identifying any obvious lane lines, sidewalks, guardrails, signal lights, traffic signs, etc. in the scene, and analyze their constraints and influences on the vehicle's subsequent driving behavior.
>     The third step is to analyze the movable objects in the scene, such as pedestrians, animals, and vehicles, and point out their relative positional relationship with the current vehicle and their possible influence on the vehicle's driving.
>     The fourth step is to give a comprehensive judgment of the vehicle's environment and classify the following situations.
>     1. Roads with unclear boundaries: Scenarios where the traversable path is ambiguous or undefined, such as rural dirt tracks, off-road trails, or roads with faded/absent markings. These severely challenge perception tasks like lane detection and drivable area segmentation.
>     2. Temporary traffic rule changes: Dynamic situations where standard traffic rules are temporarily altered by construction zones, human traffic controllers, or temporary signage, requiring autonomous vehicles to adapt to unusual instructions and road layouts.
>     3. Unconventional dynamic obstacles: Features dynamic actors or obstacles uncommon in typical urban driving that demand specialized interaction strategies. Examples include large or erratically moving vehicles, vulnerable road users in unexpected locations, or animal encounters, all posing sudden hazards.
>     4. Challenging road conditions: Encompasses scenarios where adverse road surfaces (e.g., potholes, mud, snow, ice) or environmental conditions (e.g., fog, heavy rain, low-light, glare) severely impair visibility or affect vehicle dynamics, complicating hazard perception and safe navigation.
>
>     Note that you need to analyze the current scene one by one to see if it belongs to the first five unstructured scenes. If there is a scene that meets the conditions, it must be added to the final list of Categories. Each category must be judged and the reasons for your judgment must be given. If no matching items are found after judging all four categories, you can judge that the current scene is a standard road scenario.
>
>     Your answer should be in the following format
>     {"Step1":{"Time Period": morning"|"noon"|"afternoon"|"night", "Weather": "sunny"|"cloudy"|"rainy", "Scene Description":"Your description"}, "Step2": A dict with the lane lines, sidewalks, guardrails, signal lights, traffic signs, etc. you detected as keys and their constraints and impacts on the vehicle's next driving behavior as values,"Step3": A dict with the movable objects you detected as keys and their relative position relationship with the current vehicle and their possible impacts on the vehicle's driving as values,"Step4":{"Analysis":"The judgment and analysis process of the four categories. You need to list each of the above categories one by one and make judgments and analyses. You need to use serial numbers to make your answers clearer.","Categories": a list of final classification results.}}
>     You must check and confirm that your answer does not conflict with the previous analysis stage to avoid misjudgment.
>     ```
>
>     These results demonstrate the effectiveness of our CoT approach in achieving higher accuracy for the classification task. We will consider including these findings in the revised manuscript to further support our methodology.

---

### Official Review · Reviewer_rC4B · 2025-07-02

**Rating:** 5
**Confidence:** 4

**Summary:**

This paper addresses a highly important and challenging problem in the field of autonomous driving: the performance bottleneck of models in unstructured, "long-tail" scenarios. The paper points out that a core obstacle is the lack of high-quality, large-scale, annotated datasets specifically designed for such scenarios. To this end, the paper presents a core contribution named Impromptu VLA: a meticulously curated driving dataset of over 80,000 video clips that focuses on four typical types of unstructured scenarios.

This work not only contributes a valuable data resource but, more importantly, proposes a systematic, scalable, VLM-centric data curation pipeline. The experimental section is thorough and persuasive, clearly demonstrating through extensive experiments on both closed-loop (NeuroNCAP) and open-loop (nuScenes) benchmarks that models trained with this dataset achieve significant improvements in safety, planning capability, and trajectory prediction accuracy.

In summary, this is a high-quality paper. It features a clear problem definition, an innovative methodology, robust experiments, and a well-defined contribution. Furthermore, it is very community-friendly, with a commitment to open-sourcing the data and models. Overall, I recommend this paper for acceptance.

**Dataset Code Accessibility:**

Yes

**Ethical Considerations:**

No, there are no or only very minor ethics concerns

**Final Justification:**

After reading the author's response and other reviews, I tend to accept this paper.

**Limitations Weaknesses:**

- The paper acknowledges in its conclusion that relying primarily on the Qwen2.5-VL model for annotation could introduce model-specific biases. Although comprehensive human verification largely mitigates this issue, the argument would be more persuasive if the final version or an appendix included more detailed statistics about the verification process. For instance, what was the initial acceptance rate for VLM-generated annotations? What percentage required corrections from human annotators? Such details would give readers a clearer understanding of the VLM's role and its limitations within the pipeline.
- The experiments are conducted primarily using models from the Qwen2.5-VL series. While these results are somewhat sufficient to demonstrate the dataset's effectiveness, adding an experiment with a VLM based on a different architecture (such as LLaVA or DeepSeek-VL) would be beneficial. Even a small-scale test could help prove that the gains from the Impromptu VLA dataset are generalizable and not exclusive to a specific model family.
- The paper's checklist notes that due to computational constraints, each experiment was performed only once, which is why error bars and tests for statistical significance are not reported. It would be good to briefly mention this limitation in the main text and clarify that the reported metrics are the result of a single run. Also, the computational cost should be reported in the paper.

**Strengths Contributions:**

- The "long-tail problem," or the challenge of unstructured scenarios, is a critical bottleneck for achieving L4/L5 autonomous driving. This paper directly confronts this core challenge, giving it high research value and practical significance. With the rise of VLM in autonomous driving, the need for high-quality training data has become paramount, making this research exceptionally timely.
- Comprehensive Evaluation Dimensions: The authors validate the dataset's effectiveness in both closed-loop (NeuroNCAP, which emphasizes planning and safety) and open-loop (nuScenes, which emphasizes prediction accuracy) settings, making the experimental conclusions highly reliable.
- The dataset, models, and code are publicly available, which greatly advance research in the field. For a paper centered on contributing a new resource, this commitment is essential.

---

> ### Author Rebuttal · Authors · 2025-07-30
>
> We appreciate the reviewer's insightful comments and have revised our manuscript to address the points raised.
>
> ---
> 1. **Detailed Statistics on Human Verification**
>
> We agree that providing more detailed statistics on the human verification process would enhance the persuasiveness of our argument regarding the mitigation of model-specific biases. To elaborate, the initial acceptance rate for VLM-generated annotations was approximately 81%. For the remaining 19% of annotations that were not initially accepted, 65% were ultimately removed, and 35% received human modifications. Within this 19%, the distribution of the four challenging categories was: Challenging Road Conditions (42.5%), Unconventional Dynamic Obstacles (32.0%), Temporary Traffic Rule Changes (22.9%), and Road With Unclear Boundaries (2.6%)."
>
> ---
> 2. **Generalizability Across Different VLM Architectures**
>
> The reviewer's suggestion to include experiments with a VLM based on a different architecture is valuable for demonstrating the generalizability of our Impromptu VLA dataset. We have conducted additional experiments using the Llama-3.2-11B-Vision-Instruct model. The results, as shown below, indicate that the gains from the Impromptu VLA dataset are indeed generalizable and not exclusive to the Qwen2.5-VL model family:
> | Method | L2 (m) 1s | L2 (m) 2s | L2 (m) 3s | L2 (m) Avg. |
> |---|---|---|---|---|
> | Qwen 3B Base+nuScenes | 0.14 | 0.30 | 0.58 | 0.34 |
> | Qwen 3B Base+Impromptu+nuScenes | **0.13** | **0.27** | 0.52 | **0.30** |
> | Qwen 7B Base+nuScenes | **0.13** | 0.28 | 0.55 | 0.32 |
> | Qwen 7B Base+Impromptu+nuScenes | **0.13** | **0.27** | 0.53 | **0.30** |
> | LLaVa 11B Base+nuScenes | **0.13** | 0.28 | 0.54 | 0.32 |
> | LLaVa 11B Base+Impromptu+nuScenes | **0.13** | **0.27** | **0.51** | **0.30** |
>
> ---
> 3. **Computational Constraints and Costs**
>
> We acknowledge the reviewer's point regarding the single-run nature of our experiments due to computational constraints. We will briefly mention this limitation in the main text of the paper, clarifying that the reported metrics are the result of a single run and that error bars and tests for statistical significance are not reported for this reason.
> Furthermore, we will include the computational costs in the paper as requested. For training the Qwen2.5-VL-3B-Instruct model:
> - Using 8 A100 GPUs, training on the nuScenes dataset for one epoch took approximately 1.5 hours.
> - Training with our Impromptu-VLA Dataset QA data required approximately 15 hours per epoch.
> For inference using a single A100 GPU, and leveraging sglang for acceleration:
> - Inference on the nuScenes test set took approximately 0.5 hours.
> - Inference on our Impromptu-VLA test set took approximately 15 hours.

---

### Decision · Program_Chairs · 2025-09-18

**Decision:**

Accept (poster)

**Comment:**

This paper introduces a new VLA Dataset: over 80,000 meticulously curated video clips, distilled from over 2M source clips sourced from 8 open-source large scale datasets.  This paper tries to address the "long tail problem", provides comprehensive evaluation dimensions, and conducts a comprehensive experimental evaluation. After extensive discussions between the authors and reviewers, all reviewers recognized the paper's contributions and agreed to accept it. AC hopes that all comments will be incorporated into the final version.